# Structural basis of the T4 bacteriophage primosome assembly and primer synthesis

Xiang Feng [1,3], Michelle M. Spiering [2,3], Ruda de Luna Almeida Santos[1], Stephen J. Benkovic [2] ✉ & Huilin Li [1] ✉

The T4 bacteriophage gp41 helicase and gp61 primase assemble into a primosome to couple DNA unwinding with RNA primer synthesis for DNA replication. How the primosome is assembled and how the primer length is defined are unclear. Here we report a series of cryo-EM structures of T4 primosome assembly intermediates. We show that gp41 alone is an open spiral, and ssDNA binding triggers a large-scale scissor-like conformational change that drives the ring closure and activates the helicase. Helicase activation exposes a cryptic hydrophobic surface to recruit the gp61 primase. The primase binds the helicase in a bipartite mode in which the N-terminal Zn-binding domain and the C-terminal RNA polymerase domain each contain a helicase-interacting motif that bind to separate gp41 N-terminal hairpin dimers, leading to the assembly of one primase on the helicase hexamer. Our study reveals the T4 primosome assembly process and sheds light on the RNA primer synthesis mechanism.

During DNA replication, the double-stranded DNA (dsDNA) is unwound by a ring-shaped helicase, which moves in the 3′ to 5′ direction on the leading-strand DNA in eukaryotes and in the 5′ to 3′ direction on the lagging-strand DNA in prokaryotes and bacteriophages[1,2]. Because all DNA polymerases extend only the 3′-OH of an existing oligonucleotide, an RNA primer is synthesized to initiate DNA synthesis. Primers are short RNA oligonucleotides synthesized by primases and are complementary to the parent DNA template. Primases are RNA polymerases in bacteria and bacteriophage, but are dual-functioning RNA and DNA polymerases in eukaryotes (e.g. Pol α)[3,4]. The priming reaction occurs frequently on the lagging-strand DNA in order to start each Okazaki fragment. The eukaryotic Pol α is associated with the replicative CMG helicase indirectly via the Ctf4 trimer[1,5] and synthesizes an RNA-DNA hybrid of 20 nucleotides (nt). In bacteria and phages, the primase is directly associated with the helicase and synthesizes RNA primers of various length, e.g. ~11 nt by the bacterial DnaG primase, 4 nt by the T7 gp4 primase, and 5 nt by the T4 gp61 primase. The priming mechanism is perhaps the least well understood aspect in DNA replication in all domains of life[3,4,6-8]. The primase counting or measuring capability to determine the length of the primer synthesized has been discussed hypothetically for the archaeal and eukaryotic primase complexes[9-11].

In bacteria and phages, the helicase and primase assemble into a functional complex known as a primosome to couple DNA unwinding with RNA primer synthesis. The bacterial DnaB helicase is composed of an N-terminal domain (NTD) and a C-terminal domain (CTD). The DnaB helicase assembles into a two-tiered hexameric ring where the N-tier has three-fold symmetry due to dimerization of the N-terminal helical hairpins, while the C-tier has nearly six-fold symmetry composed of six CTDs with six ATP binding pockets at the CTD-CTD interfaces[12]; and the N-tier and the C-tier are staggered in a domain-swapped fashion[12,13]. The DnaB helicase was observed in two states: a "dilated" state where three N-terminal helical hairpin dimers form an equal-sided triangle and the C-tier encircles a large channel, and a "constricted" lock-washer state where the N-terminal helical hairpin dimers swivel inward to bring the CTDs towards each other constricting the central chamber to accommodate only single-stranded DNA (ssDNA)[14,15]. The bacterial DnaB helicase is thought to translocate on ssDNA utilizing a "rotary-staircase" mechanism in which subunits translocate sequentially from one end of the "lock washer" to the next at the expense of ATP

[1]Department of Structural Biology, Van Andel Institute, Grand Rapids, MI, USA. [2]Department of Chemistry, The Pennsylvania State University, University Park, PA, USA. [3]These authors contributed equally: Xiang Feng, Michelle M. Spiering. ✉e-mail: sjb1@psu.edu; huilin.li@vai.org

hydrolysis[1,13,15]. The bacterial DnaG primase is a 65-kDa, three-domain protein with an N-terminal Zn-binding domain (ZBD), a core RNA polymerase domain (RPD), and a C-terminal helicase-interacting domain (HID)[16]. Structures are available for the individual domains, but not for any full-length primase, suggesting that these domains are flexibly connected[17–22]. It's been suggested that the DnaG primase and DnaB helicase form a primosome with a 3:6 stoichiometry mediated by the interaction between the C-terminal HID of DnaG and the N-terminal hairpin dimer of DnaB[17,23]. However, no structure has been reported for any bacterial primosome – either alone or in complex with ssDNA and/ or an RNA primer.

The T7 bacteriophage gp4 helicase belongs to the superfamily of prokaryotic replicative helicases that includes DnaB[2]. The T7 replication system is unique in that the primase and the helicase are fused into a single polypeptide, gp4, such that the stoichiometry of the T7 primosome must be 6 primases to 6 helicases. The gp4 helicase region lacks the N-tier ring and has only the C-tier ring equivalent to the bacterial DnaB helicase. Therefore, the T7 primosome has largely a six-fold symmetrical architecture. A recent cryo-EM study revealed that the T7 gp4 helicase has a right-handed spiral shape and exists in several states, suggesting that the helicase translocates on the lagging-strand DNA with a hand-over-hand "rotary-staircase" mechanism[24]. The same study also captured a state in which the ZBD of the gp4 primase region was handing over the RNA primer to the DNA polymerase. This structure revealed the coordination between the primase and the DNA polymerase but does not inform how the primer was synthesized.

T4 bacteriophage has been a classic model system for studying DNA replication and RNA priming mechanisms[6,25]. The T4 gp41 helicase also belongs to the prokaryotic replicative helicase superfamily[2]. The gp41 helicase and the gp61 primase assemble into a primosome complex to couple parental DNA unwinding with priming activity on the lagging-strand DNA[25]. The domain architecture of the gp41 helicase resembles the bacterial DnaB helicase[13,14] with an NTD and a RecA-like CTD that are connected by a linking helix (LH) (Fig. 1a). In the presence of ATP, the gp41 helicase monomers assemble into a hexamer to encircle the lagging-strand DNA[26] and move in a 5′ to 3′ direction to unwind duplex DNA at a rate of 30 bp/s[27,28]. The gp61 primase is composed of an N-terminal ZBD and a C-terminal RPD that are linked by an 18-residue flexible loop (Fig. 1a). The gp61 primase recognizes either 5′-GTT or 5′-GCT on the ssDNA template as a priming start site and synthesizes a pentaribonucleotide primer[29]. The underlying mechanism for synthesizing this length of primer by the T4 primosome is unknown.

In this study, we determined a series of structures of T4 primosome assembly intermediates and the assembled primosome in a DNA-scanning mode and in a post-RNA primer-synthesis mode. We show that one gp61 primase binds to the helicase hexamer through interacting interfaces on both the ZBD and RPD of the primase. We also explored the influence the primase linker loop has over the length of the synthesized RNA primer. In sum, the structural study has enabled us to propose a unique helicase activation mechanism, a detailed process for primosome assembly, and a plausible mechanism for pentaribonucleotide primer synthesis by the T4 primosome.

## Results

### Cryo-EM of T4 primosome assembly intermediates

We assembled the T4 primosome in vitro with purified components through a process like the one described previously[30] (Fig. 1b) and resolved the intermediate states during primosome assembly (Fig. 1c). We started with the assembly of the gp41 helicase hexamer by adding ATPγS (step 1). Our previous work showed that a 45-nt ssDNA with a priming recognition site was required for detectable priming activity[31] and that a longer ssDNA substrate might be required to accommodate a DNA loop between the helicase and primase[32]. For these reasons, we designed a 70-nt ssDNA template with a 5-nt RNA primer annealed in

the middle as a substrate for the helicase and product for the primase and added it to the helicase solution (step 2). Finally, we added the purified gp61 primase to form a minimalist T4 primosome (step 3). Each mixture was incubated for 30 min at room temperature before the reaction products were withdrawn to prepare the cryo-EM grids. Through cryo-EM analysis, we determined a total of eight 3D maps that represented three states of the primosome assembly and/or functional states (Fig. 1c, Supplementary Figs. 1–6, Supplementary Tables 1, 2).

The 3D map (map 1-I at 5.7 Å resolution) derived from step 1 showed the gp41 helicase hexamer in the absence of ssDNA as a right-handed open spiral with a 10-Å gap between subunits A and F wide enough for the passage of an ssDNA. From step 2, we derived two 3D maps of the gp41 helicase hexamer bound to the ssDNA template. One map (map 2-I at 3.5 Å resolution) showed the ssDNA template in the helicase central channel, but the helicase hexamer remained in the right-handed open spiral form. In contrast, the other map (map 2-II at 3.4 Å resolution) showed the gp41 helicase bound to the ssDNA template that had undergone major conformational changes to become a planar ring closing the gap between subunits A and F and likely represents the active helicase configuration. From step 3, we determined two additional 3D maps. One map (map 3-I at 2.9 Å resolution) showed an open spiral gp41 helicase hexamer with the ssDNA template bound inside the central channel similar to map 2-I. The other map (map 3-II at 3.2 Å overall resolution) showed the active gp41 helicase hexamer as a planar ring encircling the ssDNA template (2.7 Å local resolution) with a gp61 primase bound. The high resolution achieved for each map allowed for atomic modeling of the intermediate states during primosome assembly.

### DNA template binding induces conformational changes that activate the helicase

The first state of primosome assembly (map 1-I at 5.7 Å resolution) showed the gp41 helicase in the absence of ssDNA as a two-tiered asymmetric homo-hexamer with an N-tier and a C-tier (Fig. 1c). The NTD can be further divided into a globular subdomain followed by a helical hairpin. The NTDs of two neighboring subunits form an anti-parallel dimer via angled packing of the two helical hairpins. Three such head-to-tail dimers associate mediated by interactions between the N-terminal globular subdomains leading to the triangular appearance of the N-tier and the trimer-of-dimers architecture of the gp41 helicase. In the C-tier of the hexameric helicase, the six CTDs are arranged nearly symmetrically with five nucleotide binding pockets at the subunit interfaces occupied with ATPγS molecules. The gp41 helicase forms a right-handed open spiral with a 10 Å gap between subunits A and F. In the absence of ssDNA, subunits A and F that line the gap of the open spiral were partially mobile to perhaps facilitate the entry or exit of ssDNA from the central channel.

The second state of primosome assembly is represented by map 2-I at 3.5 Å resolution and map 3-I at 2.9 Å, each with the ssDNA template bound in the central channel but with the gp41 helicase remaining an open spiral. Initial binding of the ssDNA caused only minor changes to the open spiral structure making it slightly more compact with a narrower gap and less mobility in subunits A and F (Supplementary Fig. 1e). These minimal conformational changes indicate that the DNA-free gp41 helicase is competent for loading onto ssDNA, although a gp59 helicase loader protein is necessary for efficient helicase loading in vivo. This is very different from the DnaB helicase, which has a closed, planar conformation in the nucleotide-bound, ssDNA-free state necessitating a helicase loader (the DnaC hexamer[12,15] or loaders from bacteriophage[33]) to open the DnaB helicase ring to load onto ssDNA.

In contrast, the third state of primosome assembly is represented by map 2-II at 3.4 Å resolution and map 3-II at 2.7 Å resolution. Both showed that three major conformational changes occurred when the ssDNA template-bound, open spiral of the gp41 helicase transitioned

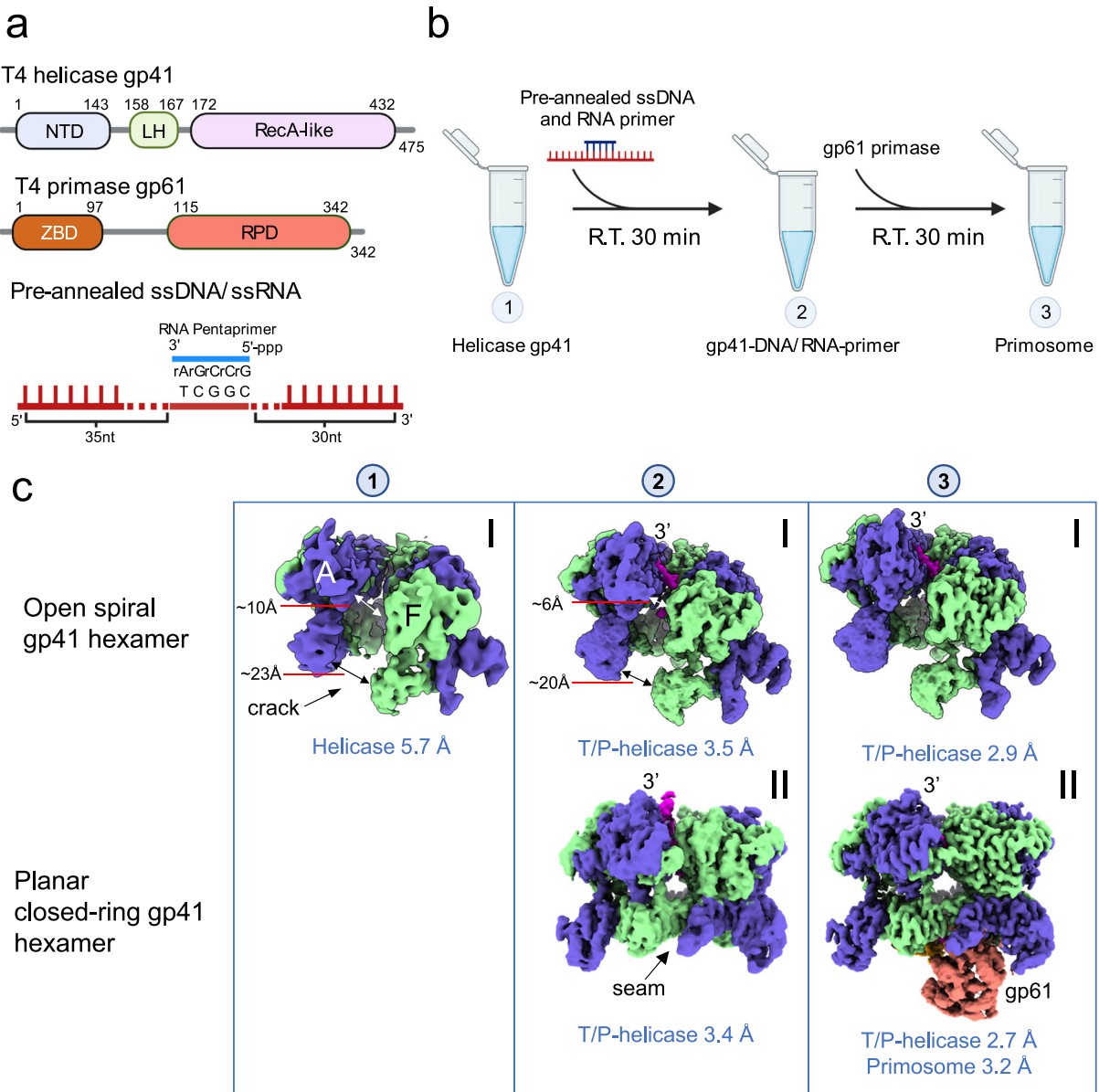

**Fig. 1 | In vitro assembly and cryo-EM analysis of the T4 primosome. a** Domain architecture of the gp41 helicase, the gp61 primase, and the ssDNA/RNA primer construct used in the in vitro assembling of the primosome. **b** Step-by-step process used to assemble the T4 primosome. **c** 3D EM maps of the step 1 complex—(I) the hexameric gp41 helicase in an open spiral configuration; two step 2 complexes—the ssDNA-bound gp41 helicase in the (I) open spiral and (II) planar closed-ring configurations; and two step 3 complexes—(I) the ssDNA-bound gp41 helicase in an open spiral form and (II) the primosome. The average resolution of each map is labeled below the respective map. The primosome map (3-II) is a composite map of the separately refined maps of the ssDNA-bound gp41 helicase (2.7 Å) and the gp61 primase region of ssDNA-bound gp41 helicase-gp61 primase complex. The overall resolution of the composite primosome map reaches 3.2 Å, but the resolution of the primase region is lower; see the detailed local resolution estimation in Supplementary Fig. 5. The EM maps are postprocessed by DeepEMhancer (v0.14, using the "tightTarget" model) and aligned in such way that the gap in open spirals or the seam in closed rings face the readers. The two open-spiral, ssDNA-bound gp41 helicase structures (2-I and 3-I) are the same; and the closed-ring gp41 helicase structure (2-II) is essentially the same as the gp41 helicase region in the primosome structure (3-II). Created with BioRender.com.

to a planar, closed-ring state (Fig. 2a, b). First, the N-tier rotated in plane by 60° with respect to the C-tier leading to a domain-swapped packing in the closed state, i.e. - each CTD now aligned with the neighboring NTD. The domain-swapped arrangement has also been observed in other members of the prokaryotic replicative helicase superfamily[12,13,15,23,34–36]. Second, the linking helix (LH) from the N-tier relocated to interact with the CTD of the neighboring subunit. Finally, the N-terminal helical hairpins (enclosed in a yellow, dashed-line shape) closed in a scissor-like motion that increased their crossing angle from 100° in the open spiral form to 165° and almost antiparallel in the closed-ring form (Fig. 2a–d). Because the gap between subunits

A and F is now closed, this map likely represents the active gp41 helicase configuration. This scissor-like activation mechanism appears to be unique to the gp41 helicase. The *E. coli* DnaB helicase was reported to use a very different aperture-like motion to close the hexameric ring and activate the DnaB helicase[15] (Fig. 2e, f).

### Active T4 helicase is a planar ring with DNA-translocating loops spiraling around the central channel

In the active gp41 helicase, five ATPγS molecules were resolved in the C-tier with the nucleotide-binding pocket between the CTDs of subunits A and F unoccupied (Fig. 2g). This is comparable to the five ADP-

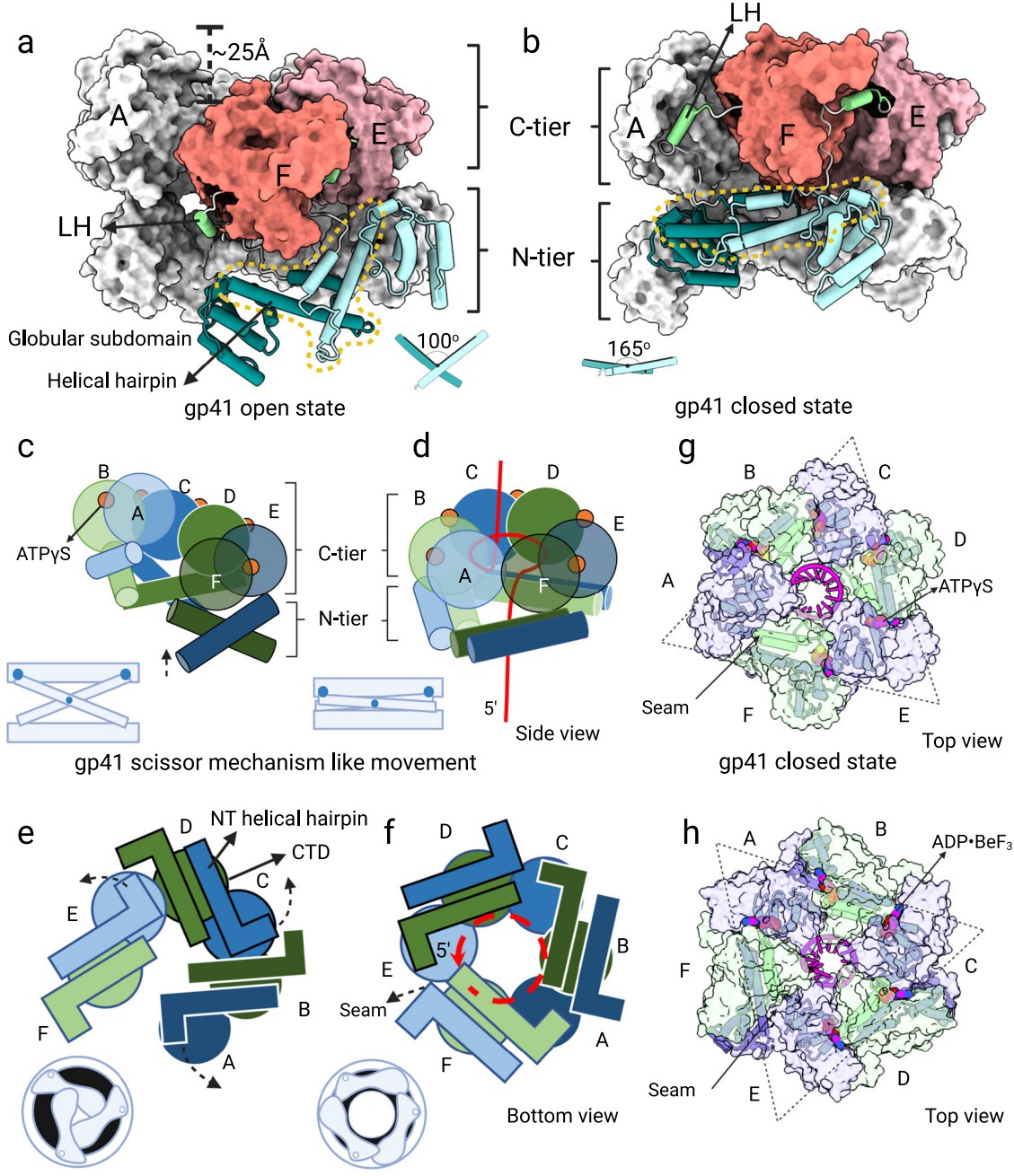

**Fig. 2 | Conformational changes from the inactive, open-spiral to the active, closed-ring structures of the ssDNA-bound gp41 helicase.** Side views of (**a**) the open spiral and (**b**) the closed-ring structure. The domains are colored as in Fig. 1a. Subunits A–D are shown as gray surfaces. The NTDs and LHs of subunits E and F are in cartoon form and their CTD are shown in colored surfaces. Illustrations depict the scissor-like motion of the gp41 helical hairpins in (**c**) the inactive, open spiral and (**d**) the active, closed-ring structures. In contrast, illustrations depict the aperture-like swivel motion of the N-terminal helical hairpins of the *E. coli* DnaB helicase in (**e**) the inactive, open spiral and (**f**) the active, closed ring (PDB entries 6QEL and 6QEM). The red curves in (**d**) and (**f**) represent the lagging-strand DNA. Top surface views of (**g**) the gp41 helicase showing the bound ssDNA and ATPγS molecules (this study) and (**h**) the DnaB helicase showing the bound ssDNA and nucleotides (PDB entry 6QEM). The dashed triangles indicate the interdimeric and intradimeric location of the seam in (**g**) the T4 helicase and (**h**) the DnaB helicase, respectively. Created with BioRender.com.

$BeF_3$ molecules observed in the DNA-bound DnaB helicase structure[15] (Fig. 2h). However, the seam (identified by the unoccupied nucleotide-binding site) in the gp41 C-tier is at the interface between the A-B and E-F dimers, whereas the seam is within the E-F dimer in the DnaB helicase. Interestingly, the active gp41 helicase is more planar, with a vertical offset between the highest and lowest subunits of only 5 Å, as compared to a 20 Å offset between equivalent subunits in the DnaB helicase (Supplementary Fig. 7).

Each gp41 helicase CTD can coordinate two ssDNA backbone phosphates with sidechains Asn327–Tyr329 of the L1 loop, Ala372–Ala375 of the L2 loop, and Lys358 (Fig. 3a). Overall, the ssDNA coils around the interior of the gp41 CTDs coordinated by the L1 and L2 loops in a similar manner in both the spiral and planar forms of the gp41 helicase; however, the ssDNA becomes more coiled as the helicase switches from the open spiral to the planar ring (Fig. 3b). In the open spiral gp41 helicase, the ssDNA binds to five (subunits B-F) of the

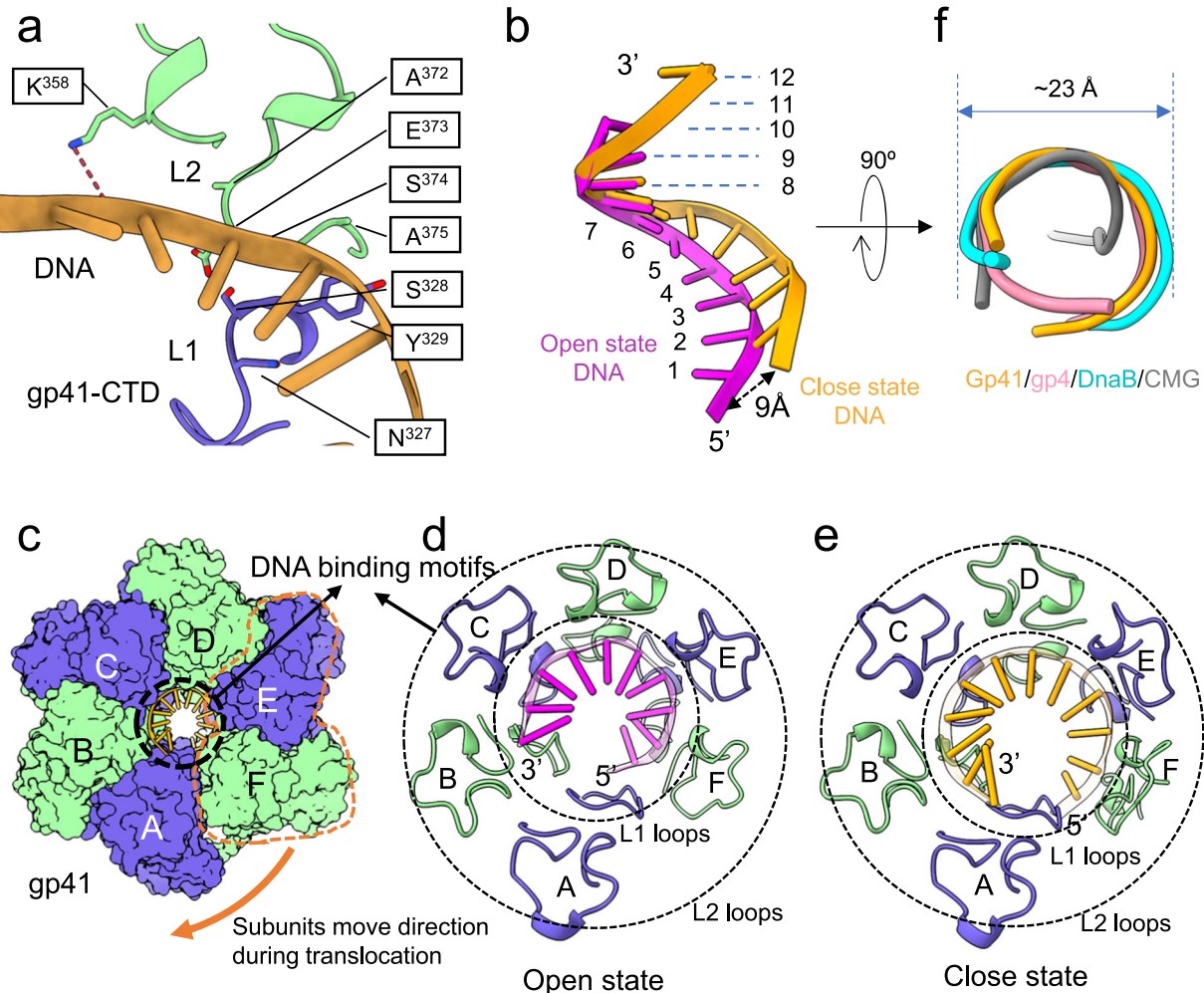

**Fig. 3 | Comparison of ssDNA binding in the inactive open spiral and active closed-ring states of the gp41 helicase. a** The ssDNA backbone is sandwiched in the gp41 CTD between the tip of the α-helix following L1 loop (blue) and L2 loop (green). **b** Superimposition of the ssDNA in the two states demonstrates that the ssDNA end moves by as much as 9 Å to adopt a flatter conformation in the closed-ring state. **c** Top view of the gp41 helicase showing the coiled ssDNA in the central channel stabilized by the gp41 CTDs. The ssDNA interacts with five gp41 CTDs in the open state (**d**), and with six gp41 CTDs in the closed state (**e**). The DNA interacting loops are between the two concentric dotted circles and shown in cartoons. The DNA bases are shown in cartoon and in sticks. **f** Superimposition of the ssDNA in the T4 helicase with ssDNA inside the T7 gp4 (PDB entry 6N9V), bacterial DnaB (PDB entry 4ESV), and yeast CMG helicase (PDB entry 5U8T). These ssDNA adopt a B-DNA-like configuration with a similar diameter of ~23 Å.

six helicase subunits and coils inside the right-handed C-tier spiral, as compared to the activated helicase ring in which the sixth subunit (subunit A) now joins the other five subunits to coordinate the ssDNA phosphate backbone (Fig. 3c–e, Supplementary Fig. 8). Therefore, the 10 nt of the ssDNA stabilized in the inactive open spiral is increased to a total of 12 nt stabilized in the active closed ring gp41 helicase. The diameter of the DNA coil is ~23 Å, similar to the dimensions of the ssDNA observed in the T7 gp4, the bacterial DnaB, and the yeast CMG helicases (Fig. 3f). Therefore, the gp41 helicase likely unwinds DNA with a mechanism similar to the T7 and DnaB helicases[20,24].

### Bipartite interaction between the primase and the helicase hexamer

The final step of primosome assembly is for the gp61 primase to bind to the gp41 helicase hexamer. Importantly, the scissor-like movement that activates the gp41 helicase exposes a hydrophobic patch on the side of the gp41 helical hairpins, which recruits the gp61 primase (Supplementary Fig. 9). Therefore, the gp41 helicase in the active closed-ring form, but not the inactive open-spiral form, can recruit a primase. The primosome map 3-II shows the gp61 primase bound to

the active, planar gp41 helicase hexamer. Interestingly, the structure of the gp41 helicase in the primosome complex is highly similar to the closed-ring helicase observed in the absence of gp61 primase (map 2-II), suggesting that the primase binding to the activated closed-ring gp41 helicase does not cause additional major conformational changes.

By classifying the primosome particles based on the gp61 primase regions and refining the entire primosome particles in each class, we obtained three primosome EM maps (Fig. 4, Supplementary Figs. 4, 5). In all three maps, the gp41 helicase has a higher local resolution of 2.7 Å than the overall primosome resolution of 3.3 Å, 3.2 Å, and 3.5 Å, respectively. This indicates the partial flexibility of the primase subunits, especially at the distal RPD region; however, sidechain level details are present at the gp41 helicase/gp61 primase interface (Supplementary Fig. 10a, b). Aided by AlphaFold2, we were able to build atomic models of a T4 primosome in the three EM maps with a resolved full-length primase. The three primosome structures are similar, but not identical, to each other because of the asymmetry of the seam present in the gp41 helicase; they differ in the location of the two gp41 helicase dimers that bind the gp61 primase (Fig. 4). The gp61

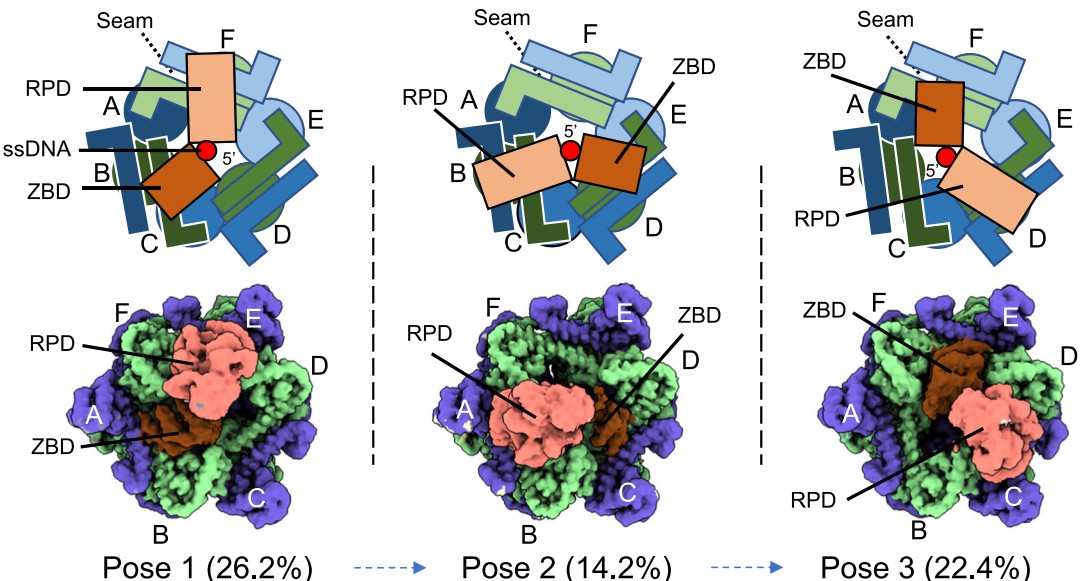

**Fig. 4 | Three gp61 primase binding poses on the gp41 helicase.** The three gp61 primase binding poses are observed in the T4 primosome in the presence of ssDNA and ATPγS. The upper panels are sketches of the three EM maps shown in the lower panels. One gp61 primase has two binding sites on the helicase hexamer: the gp61 ZBD binds to two NTDs of one gp41 dimer, and the gp61 RPD binds to two NTDs of a second gp41 dimer. The two NTDs of the third gp41 dimer are unoccupied. In each binding pose, the ssDNA 5′-end emerging from the gp41 helicase consistently passes between the gp61 ZBD and RPD domains. The particle population of each pose is given in parentheses. These poses are similar, but not identical, and are referred to as binding poses 1 through 3. The poses are non-equivalent because of the asymmetric helicase structure due to the presence of a seam between subunits A and F.

ZBD and RPD bind the gp41 helicase similarly in the three binding modes with the ZBD binding to the first N-terminal helical hairpin dimer and the RPD binding to the second helical hairpin dimer in a clockwise direction when viewed from the primase side of the primosome. The three spatial and stationary binding modes of gp61 primase may represent three temporal steps of a primosome as the gp41 helicase translocates on ssDNA.

We found that ~16% of the primosome particles had gp61 primase flexibly associated with it such that the gp61 primase density/location could not be accurately determined. In addition, a subpopulation of primosome particles (4.9%) was identified with only the gp61 RPD stably bound to the gp41 helicase, while the position of the gp61 ZBD was too disordered to be seen and was presumed to be flexible in solution (Supplementary Fig. 4). The stoichiometry of the T4 primosome has been controversial, with reports ranging from 6:1[30,37] to 6:6[31,38] helicase to primase subunits in the complex. The bipartite binding mode of gp61 primase to gp41 helicase observed in the majority of the primosome particles is consistent with a stoichiometry of 6 helicase:1 primase. But static structures can underestimated the complexity of protein quaternary structures and/or miss the dynamic equilibrium between different quaternary forms of complexes in solution[39]. The observation of subpopulations of primosome particles with other than bipartite or undeterminable primase binding modes to helicase suggests that other than 6:1 or flexible stoichiometries are also possible for the T4 primosome in solution.

In our structure, the gp61 primase interacts with the gp41 helicase in a bipartite manner with both the ZBD and the RPD (Fig. 5a, b). The gp61 ZBD interacts with the gp41 helicase via a helicase-interacting motif (HIM1; Ile74–Lys97) consisting of a helix-turn-helix motif following the Zn-ribbon core, and the RPD interacts with the gp41 helicase via a second helicase-interacting motif (HIM2; Ala327–Lys341) consisting of a single α-helix at the C-terminus following the catalytic TOPRIM fold (Fig. 5a). The gp61 HIM1 and HIM2 motifs bind to a similar region on separate gp41 NTD dimers (Fig. 5b). The gp61 HIM1 binds primarily to the gp41 helical hairpin, but also contacts the globular subdomain of the gp41 NTD (Fig. 5c). Specifically, Lys79, Glu80, Phe81, and Pro83 in the short first α-helix and the turn region of the gp61 HIM1 form Van der Waals interactions with the gp41 globular subdomain; and the longer second α-helix of the gp61 HIM1 forms an extensive interface with the gp41 helical hairpin, involving the Tyr86-Arg94 region of gp61 primase and the Phe104–Glu118 region of gp41 helicase. Sequence alignment reveals that the gp61 HIM1 is not conserved, suggesting that the interaction between the primase ZBD and the helicase may be unique to the T4 primosome (Supplementary Fig. 11). The gp61 HIM2 forms a helix-helix packing interaction with the N-terminal helical hairpin of the gp41 NTD dimer (Fig. 5d). The interface is largely hydrophobic, but also includes two hydrogen bonds - one between Lys333 of gp61 and Ser108 of gp41 and the other between Ser337 of gp61 and Thr115 of gp41. Comparison of the T4 gp61 primase and the bacterial DnaG primase reveals that the gp61 primase contains the first two domains of the DnaG primase but lacks the third domain of the bacterial primase (Supplementary Fig. 12a). The third DnaG primase domain functions to interact with the DnaB helicase and the ssDNA-binding protein[40]. Therefore, the interaction between the gp61 primase and gp41 helicase is expected to be different from the bacterial system. In contrast to the single-helix gp61 HIM2, the bacterial DnaG primase uses a C-terminal four-helix bundle as the HID to interact with DnaB[19] (Supplementary Fig. 12b, c).

In our initial primosome structure, there was no density for the RNA primer, but the density for the ssDNA was present stretching from the gp61 ZBD to the N-terminal lobe (NTL) of the gp61 RPD (Fig. 5b); therefore, this structure may represent an ssDNA-scanning configuration of the primosome in search of a priming recognition sequence. The ZBD zinc-ribbon core is highly conserved and recognizes primer start sites on the template DNA[22,41,42]. In our structure, several residues with bulky side chains (Trp53, Tyr55, His64, Tyr66, and His71) of the zinc-ribbon core interact with the bases of the ssDNA, similar to the interactions between the T7 gp4 ZBD and ssDNA (Fig. 5e, Supplementary Fig. 10c, d)[24]. In the gp61 RPD, the topoisomerase-primase fold (TOPRIM; Thr217–Ile326) is sandwiched between the α/β-folded NTL (Lys115 to Ala216) and the HIM2 motif (Ala327–Ile342) (Fig. 5a).

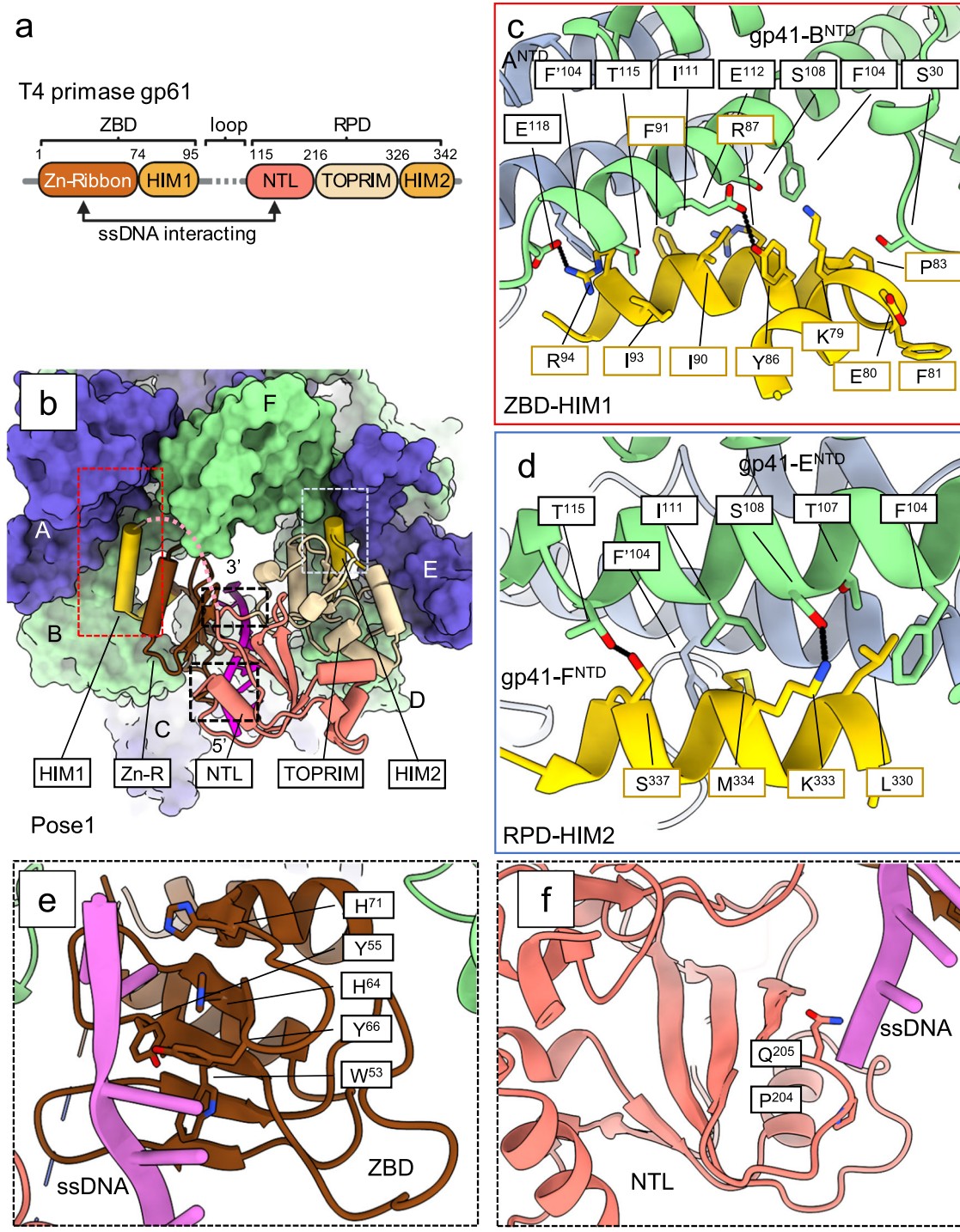

**Fig. 5 | Interaction of the gp61 primase with the active and closed-ring gp41 helicase. a** A detailed architecture of the gp61 primase indicating the NTL: N-terminal lobe of the RPD and the TOPRIM, a signature domain conserved in all primases. **b** A tilted bottom view of the T4 primosome showing the gp61 primase in cartoon binding to the NTDs of the gp41 helicase in surface. The binding mode is essentially the same in all three gp61 binding poses. The gp61 domains are colored as in (**a**). Enlarged views showing the detailed interactions between gp41 NTDs and (**c**) the HIM1 and (**d**) the HIM2 of the gp61 primase. The interacting residue pairs are connected by dashed black lines. Enlarged views showing detailed interactions between ssDNA and (**e**) the zinc-ribbon core and (**f**) the NTL in the RPD of the gp61 primase. Created with BioRender.com.

TOPRIM is the catalytic site for phosphate transfer and is conserved among the bacterial and phage primases. We found that the gp61 NTL binds the ssDNA via the edge of the β-sheet core, similar to the ssDNA binding by the bacterial DnaG primase[20]; however, the bacterial DnaG NTL is longer and contains a second DNA binding site, a β-hairpin motif, that is absent in the gp61 RPD. Nevertheless, the ssDNA in the DnaG RPD would be an extension of the ssDNA emerging from the

gp41 helicase in our primosome structure, indicating a similar ssDNA-binding mode for the bacterial and phage primases (Fig. 5f, Supplementary Fig. 10e, f).

**Possible primer length determinant**
We hypothesized that the absence of the RNA primer density in our primosome EM map was due to slow ATPγS hydrolysis by the WT gp41

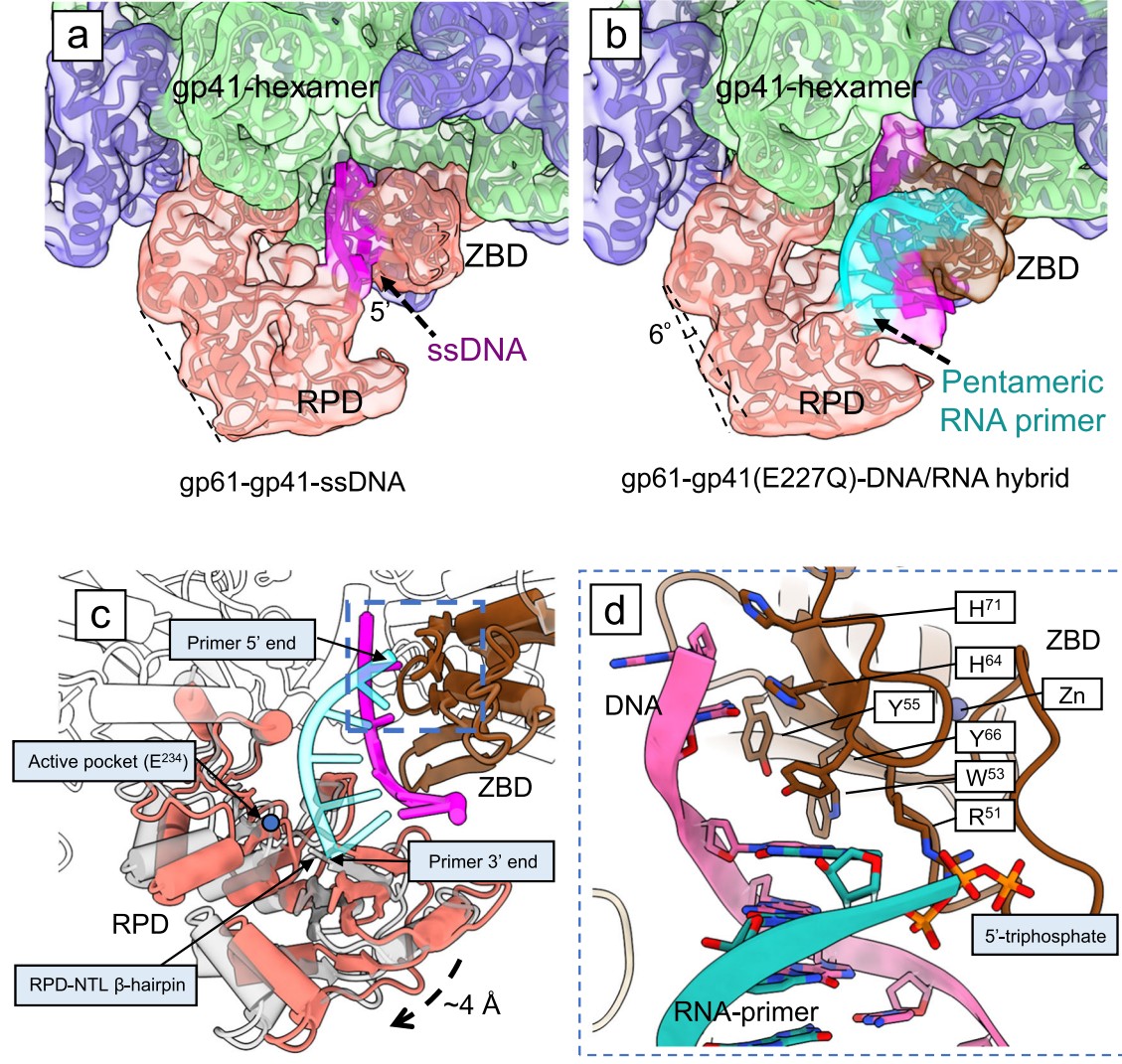

**Fig. 6 | Two conformations of the T4 primosome captured by cryo-EM.** Comparison of EM maps and models of (**a**) the WT primosome with a resolved ssDNA and (**b**) the helicase mutant (E227Q) primosome with a resolved ssDNA/RNA primer duplex. The EM maps are shown in transparent while the structural models are shown in cartoon. **c** Superimposition of the two primosome structures showing that the gp61 ZBD is stationary, while the gp61 RPD rotates outwards by 4 Å to accommodate the 5-nt RNA primer. The structures are aligned by the gp41 NTD.

Note that the 3'-end of the primer has passed the primase catalytic site—the E234 that is marked by a blue dot. The structures are colored as in (**b**) except that gp61 RPD of the mutant primosome is shown in transparent gray. **d** Enlarged view of the dashed blue box region in (**c**) showing the interactions between the gp61 ZBD and the ssDNA/RNA primer. All labeled residues interact with the DNA (magenta) except for Arg51 and the preceding loop that interact with the 5'-triphosphate group of the RNA primer (green).

helicase. Therefore, we utilized an inactive gp41 helicase mutant with an E227Q substitution (Supplementary Fig. 13) in the walker-B motif that abolished the ATP-hydrolysis activity[24], and assembled a T4 primosome with the WT gp61 primase and the ssDNA/RNA primer substrate. We separately refined the helicase and primase regions to improve the resolution (Supplementary Figs. 14, 15, Supplementary Table 3). The mutant gp41 helicase region had a resolution of 3.6 Å and revealed a structure essentially the same as the WT helicase (Supplementary Fig. 14c). The gp61 primase region achieved ~4.0 Å local resolution where the pentaribonucleotide primer was observed to form a short hybrid duplex with the template DNA that was absent in the WT primosome map (Fig. 6a, b). The RNA primer density was just enough to accommodate five ribonucleotides, consistent with the RNA primer provided. Interestingly, we found that the gp61 RPD rotated around the HIM2 by ~6° away from the ZBD to engage the ssDNA/RNA duplex, as compared to the gp61 primase in the WT primosome where the RNA primer is missing (Fig. 6a, b). Superposition of the gp61 primase structures of the WT and mutant primosomes shows that the

RPD moves outward by 4 Å to accommodate the ssDNA/RNA duplex in the mutant structure (Fig. 6c).

In the mutant primosome, rotation of the gp61 RPD caused the ssDNA template to rotate and interact more extensively with the gp61 ZBD as compared to the ssDNA in the WT primosome. The gp61 ZBD residues Trp53, Tyr55, His64, Tyr66, and His71 form multiple stacking interactions with the DNA bases (Fig. 6c, d). The 5-nt RNA primer extends from the gp61 ZBD to the NTL of the gp61 RPD. The primer 5'-triphosphate group is stabilized by Arg51 in the gp61 ZBD loop region (Fig. 6d); similar to the interaction observed in the T7 gp4 ZBD–ssDNA/RNA primer structure[24]. Therefore, the ZBD-mediated coordination of the 5'-end of the RNA primer is likely conserved among the phage priming systems. In the gp61 RPD, the RNA primer binds to the NTL and displaces the ssDNA bound there in the WT primosome structure. The 3'-end of the primer is stabilized by the β-hairpin of the gp61 RPD (Fig. 6c). Interestingly, the 3'-OH of the primer has passed over the catalytic site in the mutant structure, such that no additional ribonucleotides can be added in this configuration. This is consistent with the

predominant 5-nt length of primers synthesized by the gp61 primase. We therefore assigned the mutant T4 primosome structure as in a post-primer-synthesis state.

A working T4 primosome moves in a 5′ to 3′ direction on the lagging-strand DNA while simultaneously synthesizing an RNA primer in the opposing 3′ to 5′ direction. The gp61 RPD is physically attached to the gp41 helicase via its HIM2 and−as we have described above−can only rotate to accommodate a growing RNA primer. However, the gp61 RPD is connected by a linker loop (Lys97 to Lys115) to the stationary gp61 ZBD docked on the helicase. As the gp61 RPD rotates to synthesize and follow the elongating 3′-end of the primer, the linker loop becomes stretched, placing a limit on the range that the domain can follow the primer 3′-end. Indeed, we found that the linker loop stretched and became ordered in the mutant T4 primosome bound to a completed 5-nt RNA primer, as compared to the disordered linker loop in the ssDNA template-bound WT primosome (Supplementary Fig. 16). Therefore, the linker loop appears to contribute to the "counting" or "measuring" function that defines the primer length of the T4 primosome.

To test this hypothesis, a series of mutant primases was designed with either shortened or extended linker loops between the ZBD and RPD that included both conservative (e.g., the deletion or insertion of 1-2 residues) and substantial (e.g., the deletion or insertion of 4−5 residues) changes to the linker length, as well as the insertion of flexible (e.g., GGGGS sequence) or rigid (e.g., a repeat of the current linker sequence of PKEL) residues into the linker (Supplementary Table 4).

The RNA primer synthesized by the WT primase and four of the five linker loop mutants was predominantly a pentamer (86 ± 1%) with minor amounts of hexamer (2.7 ± 0.4%), tetramer (2.2 ± 0.4%), and trimer (9 ± 1%) RNA primers observed (Supplementary Fig. 17). Only in the case of mutant linker 2, where the linker loop was shortened by four residues, was a small but appreciable effect on the length of the primer noticed. While the predominant RNA primer was still a pentamer, the amount of pentamer (82.4 ± 0.6%) and hexamer (1.0 ± 0.1%) primer decreased slightly and the amount of trimer primer (13.9 ± 0.1%) increased with respect to the WT and other mutant primases as one might expect if the shorter linker loop restricted the rotation of the RPD thereby limiting the length of the primer that could be synthesized.

## Discussion

Our systematic structure analysis has provided insights into the assembly process of the T4 primosome and its simultaneous DNA translocating and RNA priming mechanism. Based on the intermediate structures described above, we propose a multi-step T4 primosome assembly process (Fig. 7a, Supplementary Movie 1). The assembly begins with the ATP-dependent oligomerization of gp41 helicase monomers. The helicase first assembles a right-handed hexameric open spiral with a lateral gap of 10 Å at its narrowest region. This gap likely serves as the lateral gate for the lagging-strand DNA to pass through. Existence of an open spiral structure is consistent with an earlier observation that the T4 helicase, but not the DnaB helicase, can

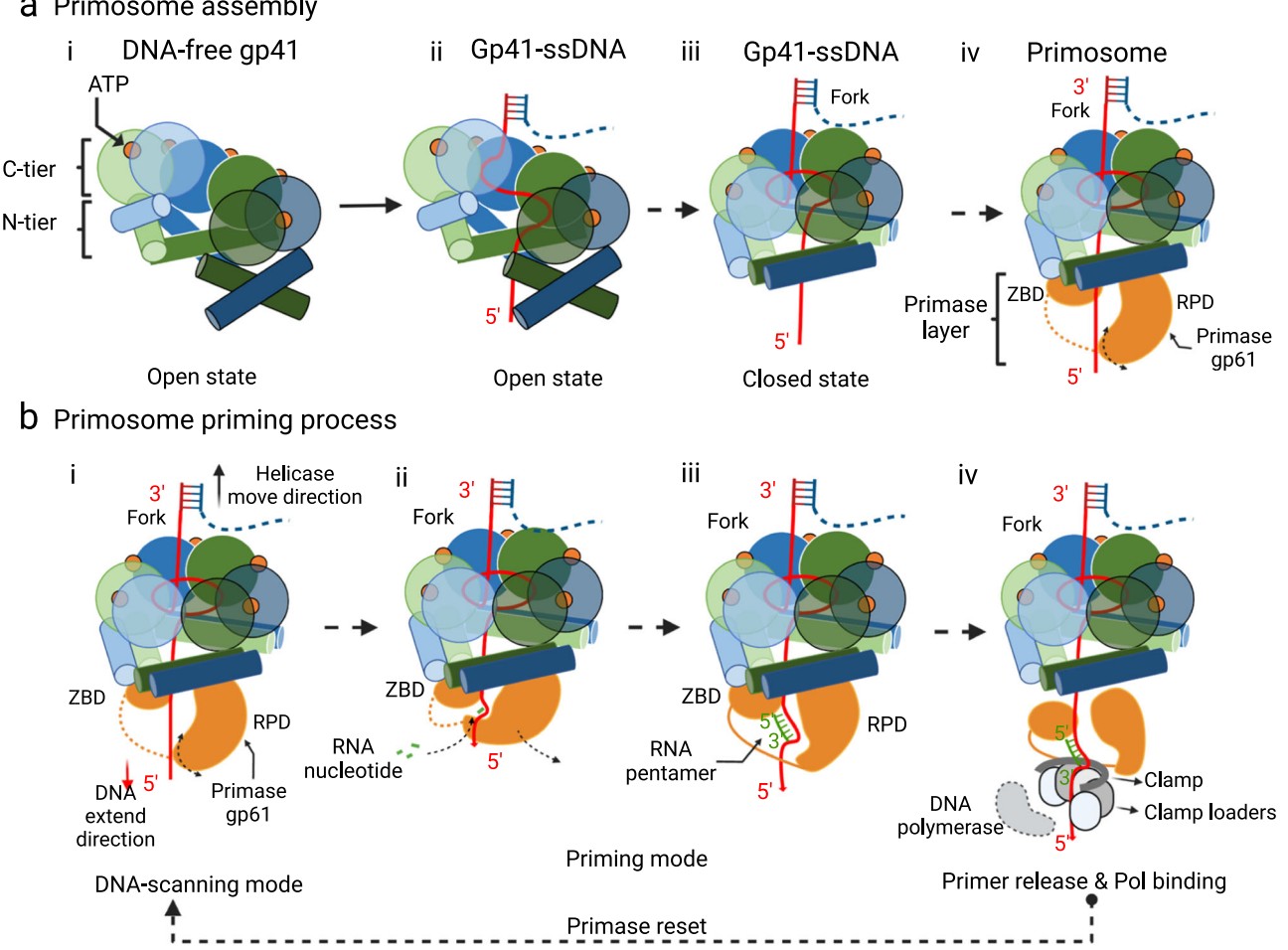

**Fig. 7 | Hypothetical model for primosome assembly and DNA-unwinding coupled RNA priming of the T4 replication system. a** The four-step primosome assembly process is based on the four observed intermediate structures in this study. Note that the helicase loader gp59 is omitted in the scheme but is required in vivo. **b** A proposed four-step DNA-unwinding coupled RNA priming mechanism (steps i to iv); see text for details. Created with BioRender.com.

self-load onto a naked ssDNA, although it does so inefficiently[43] and therefore pairs with the gp59 helicase loader for efficient DNA loading in vivo. Interestingly, ssDNA binding in the helicase central channel does not immediately induce gate closure, suggesting that the ssDNA-bound open spiral is a metastable state and that there is an energy barrier between the open spiral and the gate-closed, active ring configuration. However, transitioning from the spiral to the planar ring does not appear to require ATP hydrolysis, because only ATPγS and not ADP molecules are present in both the spiral and ring forms. Therefore, we suggest that the binding energy of the N-tier helical hairpin dimers post the "scissor-like" motion drives the inactive-to-active conformational transition. And the binding energy of the two additional DNA nucleotides inside the active helicase further stabilizes the active configuration. This binding-energy-driven transition hypothesis is supported by our observation that the mutant gp41 helicase, lacking ATP hydrolytic activity, also adopted the active planar ring in the assembled primosome. Importantly, the conformational change in the N-tier helical hairpin dimers exposes an otherwise hidden hydrophobic surface to recruit gp61 primase to assemble the primosome. It is possible that the cryptic primase-binding site in the helicase may have evolved to prevent the primase from binding until after the helicase is activated and encircles an ssDNA. Docking of a gp61 primase to the helicase hexamer completes the assembly of a functional primosome.

We have described the T4 primosome in two distinct states (Supplementary Movie 2). The first state is observed in the WT primosome in which the primase interacts with the ssDNA template in three largely equivalent poses. We suggest this is a DNA-scanning mode of the primosome in which the primase ZBD scans the ssDNA sequence for a 5′-GTT or 5′-GCT primer start site[29] (Fig. 7b, step i). Once a primer start site is detected, we expect that the gp61 catalytic RPD will swing towards the ZBD where the primer start site is bound and initiate primer synthesis (Fig. 7b, step ii). This step is hypothetical as we did not experimentally capture this state in our cryo-EM grids. Because DNA translocation by the gp41 helicase (5′ to 3′) and primer synthesis by the gp61 primase (3′ to 5′) occur in opposite directions on the lagging-strand DNA, the gp61 RPD will need to rotate away from the ZBD during primer synthesis to maintain contact with the nascent 3′-end of the RNA primer. Indeed, after the 5-nt RNA primer (in the RNA/DNA hybrid duplex) has been fully synthesized, the RPD has rotated to the furthest possible extent away from the ZBD such that the linker loop connecting the two domains appears to be stretched and becomes ordered in the cryo-EM map (Fig. 7b, step iii). We hypothesized that the linker loop length could contribute to the "measuring tape" function of the gp61 primase that limits the primer to 5 nt in the T4 system. However, shortening the linker loop by four residues had only a subtle effect on the length of the synthesized primer producing slightly less pentamer and hexamer primers and more trimer primers. Therefore, other factors besides the linker loop also likely influence the length of the primer synthesized by the T4 primase. One potential factor is the affinity of the primase to the helicase. The elongating ssDNA emerging from the gp41 helicase may apply increasing pressure on the primase to dissociate from the helicase, leading to the termination of primer synthesis. Such a scenario is consistent with the knowledge that T4 primase alone has little activity and is fully active only in complex with the helicase in the context of the primosome[44,45].

At the end of the priming reaction when a 5-nt RNA primer has been synthesized, the RNA primer/ssDNA must be handed over to the lagging-strand DNA polymerase through a mechanism that is currently unknown. Previous studies have found that handoff to the lagging-strand polymerase is stochastic and is successful between 20 and 40% of the time depending on the gp45 clamp and gp44/62 clamp loader levels[46]. We have also shown that a fraction of RNA primer/gp61 primase complex dissociates from the gp41 helicase suggesting that the helicase/primase affinity in vitro is in the micromolar range[32]. This dissociated primer/primase complex remains on the lagging-strand replication loop and can act to block further replication of the lagging strand by the polymerase producing gaps in the replicated DNA for the T4 replisome[47]. We speculate that the post RNA primer-synthesis configuration with the completed RNA primer rotated past the primase active site allows the gp44/62 clamp loader to recognize and bind the 3′-OH to initiate primer handoff by loading the gp45 clamp and gp43 DNA polymerase on to the RNA primer/ssDNA (Fig. 7b, step iv). Alternatively, the elongating ssDNA emerging from the gp41 helicase applies pressure on the completed RNA primer/primase complex to dissociate from the helicase thereby creating a signal for lagging-strand polymerase recycling. A balance between RNA primer handoff to the polymerase and RNA primer/primase signals is necessary for coordinated leading- and lagging-strand synthesis in the T4 system. Further comment on the nature of the primer handoff will require cryo-EM analysis of the T4 replisome comprised of the helicase, primase, and DNA polymerase anchored by clamp on DNA.

## Methods

### Mutagenesis, expression, and purification of the T4 proteins

The construction of expression plasmids for WT gp59 helicase loader[48], WT gp41helicase[49] and WT gp61primase[31] were described previously. The inactive gp41 helicase was generated by introducing the site-directed mutant E227Q into the gp41-intein fusion pET-IMPACT vector[49]. The mutation was made using the QuikChange Site-Directed Mutagenesis Kit (Agilent) with the mutagenic forward primer 5′-GTT CTT TAC ATT TCC ATG **C**AA ATG GCA GAA GAA GTC TG-3′ (the boldface underlined letter indicates the mutation site) and its reverse complement.

We generated a series of gp61 primase linker loop mutants (Supplementary Table 4) by introducing insertions or deletions between Proline 106 and Lysine 111 of the linker loop into the gp61-intein fusion pET-IMPACT vector[31]. The modifications were made in the middle of the linker to avoid disrupting any interactions between the ZBD or RPD and the ends of the linker. The mutations were made using the QuikChange Multi Site-Directed Mutagenesis Kit (Agilent) with the mutagenic primers in Supplementary Table 4 (the underlined portions align to the WT primase sequence; the red dashes or bases indicate deletions or insertions, respectively).

The self-cleaving, intein-based expression plasmids were transformed into *E. coli* BL21(DE3) cells and grown in NZCYM media at 37 °C to an optical density of 0.4 at 600 nm. The cultures were then cooled to 18 °C, and protein expression was induced with 0.4 mM isopropyl 1-thio-β-D-galactopyranoside. After 16–20 h of shaking, cells were harvested by centrifugation and either resuspended in chitin column binding/high salt buffer (20 mM TrisOAc pH 7.8, 1 M NaOAc, 0.1 mM EDTA, and 10% glycerol) with a cocktail of protease inhibitors (Roche) or frozen for protein purification later. Cells were lysed using sonication or high-pressure homogenization, and cell debris was pelleted at 40 000 × g. Cell-free extract was loaded onto chitin resin (New England Biolabs) for chitin-based affinity chromatography, and the chitin resin was washed with chitin column binding buffer. The resin was resuspended in cleavage/low salt buffer (20 mM TrisOAc pH 7.8, 100 mM NaOAc, 0.1 mM EDTA, and 10% glycerol) with 75 mM β-mercaptoethanol and incubated overnight at 4 °C to facilitate intein-mediated cleavage. The protein was eluted from the chitin column in low salt buffer for anion exchange (gp41 helicase) or cation exchange (gp59 helicase loader and gp61 primase) chromatography developed with a linear gradient of high salt buffer. The eluted protein was dialyzed into storage buffer (10 mM TrisOAc pH 7.8, 25 mM KOAc, 5 mM Mg(OAc)₂, 2 mM dithiothreitol, and 20% glycerol) and analyzed for purity using SDS-PAGE. Protein concentrations were determined by measuring the absorbance at 280 nm using extinction coefficients based on the protein sequence. Proteins for cryo-EM were purified in buffers containing HEPES pH 7.8 instead of TrisOAc pH 7.8 and were

frozen for storage immediately following ion exchange chromatography without dialysis into storage buffer.

## Protein activity assays

Helicase Unwinding Assay. The helicase unwinding assays were performed in triplicate in replication buffer (25 mM TrisOAc pH 7.8, 150 mM KOAc, and 10 mM Mg(OAc)$_2$) containing 50 nM unwinding fork DNA, 500 nM trap ssDNA, 2.5 mM ATP, 350 nM gp59 helicase loader, 300 nM gp61 primase, and 300 nM of either WT gp41 or gp41(E227Q) helicase in a typical reaction volume of 60 μL. The reaction was carried out at 37 °C; 10 μL aliquots were withdrawn at various time points over 5 min and quenched with an equal volume of loading buffer (240 mM EDTA, 0.2% SDS, 15% glycerol, 1 μg/mL bromophenol blue, and 1 μg/mL xylene cyanol FF). Reaction products were separated by 10% native PAGE in 1x TBE buffer and analyzed using a phosphorimager. The sequences of the oligonucleotide substrates were as follows: fork lead (5′-CAT CAT GCA GGA CAG TCG GAT CGC ATG CAG ATT TAC TGT GTC ATA TAG TAC GTG ATT CAG-3′); fork lag (5′-TAA CGT ATT CAA GAT ACC TCG TAC TCT GTA CTG CAT CGC ATC CGA CTG TCC TGC ATG ATG-3′); and trap (5′-CTG CAT GCG ATC CGA CTG TCC TGC ATG ATG-3′). The unwinding fork was made by mixing fork lead and fork lag DNA in equal molar amounts and radiolabeling the 5′-ends with T4 Polynucleotide Kinase and [γ-$^{32}$P]ATP. The identity of the bands on the native gel were confirmed by separately radiolabeling the fork lead and fork lag oligos and running them in separate lanes along side of the timepoints from the unwinding reactions.

Primase Priming Assay. Priming reactions were performed in triplicate in replication buffer containing 1.5 μM ssDNA oligo (71-mer), 4 mM ATP, 200 μM each CTP, GTP, and UTP, 4 μCi of [α-$^{32}$P]CTP, 3.0 μM gp59 helicase loader, 3.0 μM gp41 helicase, and 1.5 μM WT or mutant gp61 primase in a typical reaction volume of 25 μL. The reactions were carried out at 37 °C; 5 μL aliquots were withdrawn at 1 min time points over 4 min and quenched with 15 μL loading buffer (167 mM EDTA, 67% formamide, and 1 μg/mL xylene cyanol FF). Priming products were separated by 20% denaturing urea-PAGE in 1x TBE buffer and analyzed using a phosphorimager. The sequence of the ssDNA oligonucleotide substrate was 5′-AGA GGG AGA TTT AGA TGA GAT GAT TGA GGA TGG AGA TGT TGA TGG AGA GAT GAT GAA TGA TGA GAT GAG GG-3′ and contained a GTT priming recognition site (underlined). The band corresponding to the pentamer primer was identified by radiolabeling the 5′-end of the synthetic RNA oligo (5′-rArCrArUrC-3′) with T4 Polynucleotide Kinase and [γ−32P]ATP; this synthetic monophosphate RNA oligo does not migrate on the denaturing gel as fast as the authentic pentamer primer synthesized by the T4 primase because it lacks the triphosphate moiety.

## DNA and RNA constructs

All DNA oligos were purchased from Integrated DNA Technologies and either PAGE or HPLC purified. A ssDNA oligo 5′-GAA TGA GGA GTA GTA GTG AAT GTA GTG AGG TAA TAT CGG CTG GTC TGG TCT GTG CCA AGT TGC TGCAAA A-3′ containing a GCT priming recognition site (underlined) was used in cryo-EM. The corresponding RNA primer 5′-ppp-rGrCrCrGrA-3′ with a 5′-triphosphate moiety was synthesized using a standard run-off transcription protocol with the T7 RNA polymerase as previously described[47]. The ssDNA and RNA primer were annealed together by heating to 90 °C and slowly cooling to 4 °C at the final concentration of 250 μM.

## In vitro assembly of the T4 primosome

We assembled the T4 primosome step-by-step with purified components as previously described[30]. We first exchanged the stocks of gp41 helicase and gp61 primase to the in vitro reaction buffer (20 mM HEPES pH 7.8, 100 mM NaCl, 10 mM MgCl$_2$ and 2 mM DTT) by centrifugal filtering with Amicon Ultra-0.5 devices (10 kDa cutoff). We started with the assembly of 6 μM gp41 helicase by adding 5 mM ATPγS (step 1). We

next added 12 μM ssDNA/RNA primer substrate (step 2). In step 3, we added 3 μM gp61 primase to assemble the primosome. The mixtures were incubated for 30 min at room temperature between steps before the reaction products were withdrawn for cryo-EM grid preparation.

## Cryo-EM grid preparation and data collection

To alleviate any preferred orientation of the particles, 0.2% n-Octyl-β-D-glucoside was quickly mixed with the various primosome assembly samples before preparing the grids. Aliquots (4 μL) of the primosome assembly samples were applied to glow-discharged holey carbon grids (Quantifoil R2/1 Copper, 300 mesh) in the climate-controlled chamber of an FEI Vitrobot Mark IV. The EM grids were blotted for 3 s with filter paper and then plunged into liquid ethane and stored in liquid nitrogen. Pilot datasets of around 300 micrographs were collected on a 200-kV FEI Arctica electron microscope equipped with a K2 summit camera (Gatan) for screening purposes. The 3D reconstruction and refinement led to preliminary 3D maps with resolutions around 6 Å to confirm the quality of the grids. Two individual datasets for each primosome assembly sample were then collected using SerialEM[50] on a TFS Titan Krios electron microscope operated at 300 kV and at a nominal magnification of 130,000× equipped with a K3 summit camera (Gatan) using the objective lens defocus range of −1.0 to −2.0 μm. All the EM images were recorded in the super-resolution counting and movie mode with a dose rate of 0.88 electrons per Å$^2$ per frame; a total of 75 frames were recorded in each movie micrograph.

## Image processing

The gp41-helicase alone dataset contained 150 micrographs for screening purposes. A dataset of 50,003 particles was selected after 3D classification and 3D refined, and polished to the final map with 5.7 Å resolution.

The gp41 helicase-DNA/RNA primer and primosome datasets, containing 4091 and 11476 movie stacks, respectively, were processed in the same fashion. The movies were drift-corrected with electron-dose weighting and two-fold binned using MotionCor2-1.4.0[51]. The contrast transfer function parameters were estimated, and the effect corrected for each micrograph with CTFFIND 4.1.10 [52]. The full datasets were split into subsets with around 2000 movie stacks and imported into Relion-3[53]. In each dataset the particles were auto-picked based on templates from the scanning results and extracted with 4-fold binning. The auto-picked particles were then imported into Cryosparc2 (v3.2)[54] for 2D classification. The "good" 2D class averages with defined structural features were selected as input for ab initio 3D model reconstruction. From this stage, the particles were automatically classified into open states and closed states for separate refinement. For each state, the particle images from the best 3D maps reconstructed from subset of the data were merged and converted to the RELION format using UCSF PyEM (https://github.com/asarnow/pyem). At this stage, the gp41 helicase-DNA/RNA primer dataset had 783,172 and 406,763 particles in the open and closed states, respectively. The gp41 helicase-gp61 primase-DNA/RNA primer dataset had 1,904,089 and 1,920,895 particles in the open and closed states, respectively. For each state, the re-extracted original-scale particles were filtered through another round of 3D classification using RELION3.1 before 3D refinement. Then the particles were further CTF-refined, polished, 3D refined again and post-processed to reach the final density map. To resolve the gp61 primase density map, another round of heterogeneous classification using Cryosparc2 (v3.2)[54] was run on the particles that belonged to the closed gp41 helicase in the gp41 helicase-gp61 primase-DNA/RNA primer datasets revealing one class of gp61 primase binding to gp41 helicase. This density map provided the mask to extract only the gp61 primase signal in all of the closed-state particles through RELION3.1. Then 3D classification without alignment was applied to the extracted partial particles revealing three major binding orientations of the gp61 primase (pose 1, 2 and 3) and a minor single

gp61 RPD binding mode (pose 4). Lastly, the partial particles from each orientation were reverted to the full particles, followed by 3D classification to select the particles that give the best results. The final density maps were generated after another 3D refinement.

To estimate the number of particles with bound primase, we set the closed-ring helicase hexamer particles required for primase binding as 100%, which were classified into three classes; two of which had EM density for the primase (84% combined) and one class had no EM density for the primase (16%). From the 84% particles with a bound primase, we further separated these particles into three primase poses (1 – 3) (26.2%, 14.2% and 22.4%, respectively) and additional primase with density only for the RPD domain (4.9%). Combined, these classes accounted for 67.7% in the 84% of the particles with electron density for primase. Therefore, the remaining 16.3% of the particles must have primase flexibly bound to the helicase hexamer such that no specific orientation could be determined.

The mutant primosome was processed similarly. A total of 10,224 movies were collected. We first obtained a subset of 340,403 particle images with the helicase in the closed state, and then used the wild-type gp41 helicase-gp61 primase structure as a template to perform another round of 3D classification, leading to a mutant primosome EM map that reached an overall resolution of 3.9 Å. Close inspection of the map revealed that the template DNA density inside the mutant helicase was discontinuous, indicative of the presence of multiple helicase poses that likely resemble those observed in the wild-type helicase. We did not perform additional 3D classification because the dataset was already small, and our focus here was to capture the ssDNA/RNA primer bound to the primase. A focus refinement with Cryosparc2 (v.32)[54] lead to a 4.1-Å resolution local map of primase/DNA-RNA hybrid. Then from the same pool of particles and using the gp41 helicase as a mask, we obtained an EM map for mutant gp41 helicase in the closed state at an overall resolution of 3.6 Å. This averaged structure is similar to the WT gp41 helicase. Both maps were post-processed with DeepEMhancer (v0.14)[55] with the wideTarget model and combined to generate a composite map using ChimeraX[56].

The resolution of all final maps was calculated based on the 0.143 threshold of the gold standard Fourier shell correlation between the two independently constructed "half" maps, with each map using half of the dataset. The local resolution maps were calculated using RELION3.1 and displayed using UCSF ChimeraX.

### Model building and refinement

The high-resolution maps (better than 3.5 Å) were post-processed with DeepEMhancer (v0.14) with the tightTarget model for better density map details[55]. The initial atomic model for the gp41 helicase was built with comparative modeling while fitting into the electron density map with the Rosetta suite[57] using the template of the DnaB helicase (PDB entry 6QEM). All the atomic models were then either manually rebuilt (like unfit regions, ligands and the ssDNA chain) or refined with the program COOT[58] followed by real-space refinement in the PHENIX program[59]. The structures were further manually refined in COOT[58] until no Ramachandran outliers could be identified. Finally, the atomic model was validated using MolProbity[60,61]. Because each state (open or closed) of the gp41 helicase in either the gp41 helicase-ssDNA or the gp41 helicase-gp61 primase-ssDNA datasets have similar conformations, the map with the best resolution for each state was used to build the model. The crossing angle between the gp41 NTD helical hairpins was calculated in Pymol (The PyMOL Molecular Graphics System, version 1.8.x; Schrödinger). The EM map of the ATPγS-bound, DNA-free gp41 helicase had a moderate resolution of 5.7 Å. The structural models of the open-state DNA-bound gp41 helicase subunits were then fit into the density map using UCSF ChimeraX[56] and refined in the PHENIX program. The initial structural model of the gp61 primase was predicted with AlphaFold[62]. Due to the flexible linker loop hindering the prediction of the domain organization, the structural models of the gp61 ZBD and RPD were individually docked into the density map and refined with Phenix. The ssDNA/RNA primer binding conformation were modeled based on the structure of the T7 gp4 (PDB entry 6N9U) and refined within the density map. The structural model alignments and figures were generated using UCSF ChimeraX[56]. The sketches in Figs. 1a, b, 2c–f, 5a, 7 were created using Biorender.com.

### Reporting summary

Further information on research design is available in the Nature Portfolio Reporting Summary linked to this article.

### Data availability

The atomic coordinates of the open spiral of the gp41 helicase hexamer generated in this study have been deposited in the protein data bank under accession code 8DUO. The atomic coordinates of the closed ring of the gp41 helicase hexamer bound to ssDNA generated in this study have been deposited in the protein data bank under accession code 8DTP. The atomic coordinates of the open spiral of the gp41 helicase hexamer bound to ssDNA generated in this study have been deposited in the protein data bank under accession code 8DUE. The atomic coordinates of the T4 primosome in pose 1, pose 2, and pose 3 generated in this study have been deposited in the protein data bank under accession codes 8DVF, 8DVI, and 8DW6, respectively. The atomic coordinates of the helicase region, primase region, and the whole primosome of a mutant T4 primosome bound to an RNA primer/DNA hybrid in a post RNA primer-synthesis state generated in this study have been deposited in the protein data bank under accession codes 8G0Z, 8DWJ, and 8GAO respectively. The 3D EM map1-I of the open spiral of the gp41 helicase hexamer generated in this study has been deposited in the EM data bank under accession code EMD-27724. The 3D EM map 2-I and map 3-I of the open spiral of the gp41 helicase hexamer bound to ssDNA generated in this study have been deposited in the EM data bank under accession codes EMD-27720 and EMD-27719. The 3D EM maps 2-II and map 3-II of the helicase region of the closed ring of the gp41 helicase hexamer bound to ssDNA generated in this study have been deposited in the EM data bank under accession codes EMD-27708 and EMD-27707. The 3D EM maps of the WT T4 primosome bound to ssDNA in poses 1, 2, and 3 generated in this study have been deposited in the EM data bank under accession codes EMD-27737, EMD-27739, and EMD-27751, respectively. The 3D EM maps of the helicase region, primase region and a composite map of the mutant T4 primosome bound to an RNA primer/DNA hybrid generated in this study have been deposited in the EM data bank under accession codes EMDB-27707, EMDB-29744, and EMD-29902, respectively. The local map refined in this study from EM map EMDB-27756 have been deposited in the EM data bank under accession code EMDB-29744. This study also used the *E. coli* DnaB helicase structure in the inactive, open spiral form (PDB entry 6QEL and in the active, closed ring form (PDB entry 6QEM), the T7 gp4 structures (PDB entries 6N9V and 6N9U), the bacterial DnaB structure (PDB entry 4ESV), and the yeast CMG helicase structure (PDB entry 5U8T). Source data are provided with this paper.

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

## Acknowledgements

Cryo-EM images were collected at the David Van Andel Advanced Cryo-Electron Microscopy Suite in the Van Andel Research Institute. We thank Gongpu Zhao and Xing Meng for facilitating cryo-EM data collection. This work was supported by the US National Institutes of Health grants GM013306 (to S.J.B.) and GM131754 (to H.L.), and the Van Andel Institute (to H.L.).

## Author contributions

X.F., M.M.S., S.J.B., and H.L. conceived and designed experiments. X.F., M.M.S., and R.L.S. performed experiments. All authors analyzed the data and participated in the manuscript preparation.

## Competing interests

The authors declare no competing interests.
