## [Peer Review File · Nature Communications]

Structural basis of the T4 bacteriophage primosome assembly and primer synthesisREVIEWER COMMENTS

Reviewer #1 (Remarks to the Author):

The manuscript from Feng et al describes a series of structures of the T4 phage primosome. They describe the mechanism by which the helicase goes from a spiral state into the planar ring necessary for translocation along DNA. They also describe structures that define the states prior to and after primer synthesis.

The paper represents a major advancement and should be published in Nat Comm. I only have a few concerns.

Primary comments

- The structures seem to have reasonable statistics except that the number of residues in the Ramachandran Allowed region is suspiciously high. Assuming that these stats are coming from MolProbity, then it is expected (in an "ideal" structure) that ~2% of residues will be in the Allowed region. At resolutions such as these, I'd expect ~5% in the allowed regime, so I'm surprised to see over twice that for most of the structures. The mutant primosome is the only structure with a typical distribution of residues in each of the regions of Ramachandran space. Additionally, the materials and methods could be more explicit with regards to the refinement of the structural models. What types of restraints were used? This is especially important for model built into the 5.7Å resolution map where fitting and refinement choices are very important for obtaining a reasonable model.

- I'm not completely sold on the mechanism for determining primer length. The authors point to a possible structural mechanism that is based on the structures. While this hypothesis is reasonable, without testing it (i.e. altering linker length and observing altered primer length), it is still a hypothesis. Some of the language in the manuscript (especially the abstract) is too strong. The proposed tape-measure mechanism is intriguing, and should be explored more thoroughly. For example, if the hypothesis is correct then it would mean that shortening the linker would shorten the primer. However, this explanation is not clear because some of the sequences in Supplemental Figure 11 show linkers that are both longer and shorter than T4. However, the T4 primer is already quite short (5bp) and it is hard to imagine how it could get significantly shorter than 4bp, the length in T7 phage. What is the linker length in T7? Additionally, the Aquifex linker is by far the shortest of the sequences in Supp Fig11. (Note: I'm assuming that the sequence labeled Aaeo_DnaG is Aquifex aeolicus. This should be clarified in the ms.) It is hard to imagine that the primer of Aquifex is much shorter than in T4, especially considering the normal growth temperature of this thermophile. This would argue against the proposed mechanism. More exploration of how this proposed mechanism fits into the broader context of phage and bacterial helicase/primase systems would be nice.

Minor comments:

- 1) Figure 5d has hydrogen bonds that are hard to see versus the atoms.
- 2) It was hard to find the linker helix and the red dashed box in figure Fig 2. Even when searching for them it took me a while to find them for some reason.
- 3) In the discussion, the authors write: "Indeed, the "scissor-like" motion into the nearly antiparallel 4-helical bundle configuration increases the interface and promotes compaction of the N-tier hairpin dimers. This binding energy driven-activation hypothesis is further supported by our observation that the mutant gp41 helicase, lacking ATP hydrolytic activity, adopted the active planar ring in the assembled primosome." I'm not following the logic between these two sentences. Please consider revising to make this connection more clear between ATPase activity and the scissoring of the helical bundles.

Reviewer #2 (Remarks to the Author):

In the manuscript by Madru et al., the authors present multiple cryo-EM structures showing how the T4 bacteriophage gp41 helicase interacts with ssDNA and the gp61 primase to form a 'primosome' complex. The structures are used to propose a molecular mechanism for primosome assembly, describe how the stoichiometry of helicase to primase is determined, and explain why RNA primers are limited to pentaribonucleotides in the T4 system. This manuscript provides several novel insights into the T4 primosome and compares the structures and proposed mechanisms with the equivalent prokaryotic/bacteriophage replication machinery. Overall, the work is clearly described and nicely illustrated, the reporting of the data and methodology are sufficiently detailed and transparent, and the structural work appears sound, however, the paper would benefit from experimental validation of some of the key conclusions.

My primary concern is that the study is purely structural, with no attempt to validate the hypotheses or mechanistic models proposed with complementary assays. In the clearest example, a major claim of the manuscript is that one of the structures—of the primase in the helicase mutant (E227Q) primosome—enables the authors to define the mechanism by which the RNA primer is limited to a pentaribonucleotide. The proposed mechanism is based on the length of the linker loop between the gp61 ZBD and RPD limiting the size of the primer, by limiting the movement of the RPD domain relative to the helicase/ZBD. Whilst this is an appealing model, the structural data alone is currently insufficient to support this conclusion. There is a clear improvement in the density for the linker in the mutant structure relative to the wild-type equivalent, additional RNA density can be seen for the RNA-DNA duplex and the RPD appears to shift slightly; however, the resolution of the maps around these key elements is limited ($\sim 5\text{-}7$ Å or worse) and the density in some parts is weak (this is improved in the DeepEMhancer maps, but the fit of the RNA-DNA duplex into this map is relatively poor). Therefore, the exact positions of the RNA-DNA hybrid, surrounding residues and linker cannot be built with complete confidence. It is also notable that the linker that has currently been built into the model for the mutant primase does not appear to be fully extended. Given the above, the manuscript would benefit significantly from additional evidence to validate the proposed model, importance of the linker length and the relevance of any amino acids/interactions that are proposed to be key for determining the RNA primer length. For example, have the authors investigated whether the linker length can be modified and thereby change the length of the primers generated in a biochemical assay?

Minor points

It is unclear why the authors chose a substrate that is centrally primed, with 30 and 35 nt of ssDNA on either side. Given the DNA seen spanning the helicase channel is only 12 nt, is it possible that the helicase could bind to the DNA with the primer either in front or behind, complicating interpretation of the data? Further explanation of the design of the experiment would aid interpretation.

The authors appear to only detect one state of the primase bound to the mutant helicase complex, rather than the three states observed for the WT complex in the presence of ATP γ S, does this tell us anything about primase recruitment or helicase translocation? For example, in the absence of ATP hydrolysis, the binding state may represent initial primase recruitment prior to any helicase translocation. Is the primase found binding across the helicase seam, where it may stabilize the closed form?

The deposited structure of the mutant helicase complex (PDB: 8DWJ) only includes the primase and DNA-RNA duplex but lacks the entire helicase. Given the structure is based on a map of the entire complex, it would be much more useful to other researchers to deposit a structure for that complete complex. Presumably, completing this model would be relatively trivial and would be more informative. Additionally, Table 1 includes the global resolution of 3.9 Å for the entire map, but this is

not the resolution that represents the density for which the model of the primase structure was built.

Page 7, final paragraph "We found that about 18% of primosome particles had gp61 primase flexibly associated..." does this refer to the "16%" class shown in Sup Fig. 4? If so, the numbers should be consistent. It is also unclear from the current manuscript how the authors know that these particles are primase associated?

Page 9, "The gp61 primase region achieved ~ 4.0 Å local resolution" – this appears to conflict with the local resolution maps displayed in Sup. Fig. 14d which suggests 5-7 Å and possibly lower resolution. This figure should also be amended so that the resolution colour scale extends beyond the lowest resolution limit so that the true limits of the resolution can be evaluated.

Page 11, top "we suggest that that the transition is driven by the binding energy of the two additional DNA nucleotides from 10 nt bound in the inactive spiral form, to 12 nt bound in the active ring." Is it possible to form the closed ring on shorter DNA constructs to test this (perhaps with the helicase-dead mutant)? Negative stain EM may be sufficient for distinguishing between open and closed on different length ssDNA substrates?

The parameters used for DeepEMhancer map modification should be provided in the methods section and where DeepEMhancer maps are used in figures, this should be noted in the figure legends.

Fig. 1c legend – The resolution value (3.2 Å) given for primase in the composite map of structure 3-II is misleading, that estimate is largely based on the helicase portion that is not shown. The local resolution of the primase region being described is significantly lower than 3.2 Å.

Fig. 6a/b – Transparent surfaces shown with the fitted model would better allow interpretation of the model and fit to the density.

Fig. 7. To avoid misleading a non-expert reader, it may help to highlight in the figure and/or legend that helicase loading would normally be performed by the helicase loader gp59 in vivo.

Reviewer #3 (Remarks to the Author):

The work by Feng et al. presents several cryo-EM structures of primosome complex formation from the T4 phage system. Using snapshots of a sequential series of structural intermediates, the authors develop a molecular framework for describing the formation of an active helicase/primase complex bound to a primer-template DNA intermediate. The structures show how DNA induces conformational changes that lead to the closing of the gp41 helicase ring. This closure exposes a cryptic hydrophobic site, consequently leading to the recruitment of the primase.

Overall, the work is of high technical quality and the proposed mechanism provides fresh and long-awaited insights into the ingenious control mechanisms that regulate both helicase-DNA interactions and primase/helicase complex formation. The researchers also provide a intriguing and plausible hypothesis about how primer length is controlled based on the observation of two different gp61 conformations and the ability of a linker loop that connects the ZBD and the RPD of the primase to stretch to a predefined limit. Given its breadth and significance, the work should be published in Nature Communications pending some minor revisions.

Primary comments

- The authors hypothesize that the physical constraint of the linker loop length between the RPD and ZBD of gp61 primase dictates the primase counting. It follows that changing this linker length can influence the size of the primer synthesized. It would be important to test biochemically whether

changing the linker length indeed affects primer length.

- The authors observe a 6:1 stoichiometry for the helicase: primase complex, which (as they note) has been suggested before. However, other studies have reported evidence for a greater number of primases binding to the helicase (e.g., Yang et al, JBC, 2005). Since three equivalent binding sites are available for the primase to bind the helicase, can the authors comment on the structural basis for their observed 6:1 stoichiometry and why it would appear to discount the existence of multi-primase binding events?
- If one aligns on the axis of the bound DNA, then either the gp41 subunits must adopt a spiral configuration to track along the backbone (which appears to have a helical rise of some 25-30 Å or so), or the L1/L2 loops must adjust their vertical position to maintain comparable contacts with the substrate. Which is it? The answer bears on whether it makes sense to call the closed gp41 ring planar, or whether this is a matter of how the helicase ring is tilted in the perspective of the figures with respect to the bound DNA.

Minor points

- Figs. 3f and 3d are called out before Figs. 3a-c. Please reorder panels and adjust callouts.
- Fig. 6d and associated text – Lys51 appears to be an Arg in the figure? Please clarify.
- Fig. SF10c-e. Please show the ssDNA and ssRNA as sticks rather than cartoon to aid in visualizing the fit to density.

Point-by-point responses to reviewer' comments

Reviewer #1 (Remarks to the Author):

The manuscript from Feng et al describes a series of structures of the T4 phage primosome. They describe the mechanism by which the helicase goes from a spiral state into the planar ring necessary for translocation along DNA. They also describe structures that define the states prior to and after primer synthesis.

The paper represents a major advancement and should be published in Nat Comm. I only have a few concerns.

We thank the reviewer for their kind comments.

Primary comments

- The structures seem to have reasonable statistics except that the number of residues in the Ramachandran Allowed region is suspiciously high. Assuming that these stats are coming from MolProbity, then it is expected (in an "ideal" structure) that ~2% of residues will be in the Allowed region. At resolutions such as these, I'd expect ~5% in the allowed regime, so I'm surprised to see over twice that for most of the structures. The mutant primosome is the only structure with a typical distribution of residues in each of the regions of Ramachandran space. Additionally, the materials and methods could be more explicit with regards to the refinement of the structural models. What types of restraints were used? This is especially important for model built into the 5.7Å resolution map where fitting and refinement choices are very important for obtaining a reasonable model.

We agree with the reviewer and are grateful to the reviewers for pointing out such issues. In the previous submitted models, we refined both structures in the Phenix and adjusted in the COOT to the point where no outliers in the Ramachandra plot could be detected **by Phenix** and the automatic refinement no longer improve the statistics on side chain conformations or Ramachandra plot.

In the revised version, we have carefully inspected all deposited structural models and manually adjusted them in the COOT to the point that almost no Ramachandran outliers can be detected by COOT. It seems that COOT is stricter on the residue Ψ - Φ distribution and has improved the statistics on Ramachandran plot. All models, except for the one with medium resolution (5.7 Å), now have higher than 94% in the "Favored" region and ~5% in the "Allowed" region. The main improvement comes from conformations of residues in the loop regions. Please find the modification in the revised Supplementary Table 1. The improved statistics are within those of published structures at comparable resolutions.

For fit the 5.7-Å resolution EM map of the DNA-free helicase, the structure of each subunit in the open-state DNA-bound helicase were fitted into the density map; then the entire structural model was further refined by Phenix using the density map as a constraint. The process is now explained in the revised Method section.

- I'm not completely sold on the mechanism for determining primer length. The authors point to a possible structural mechanism that is based on the structures. While this hypothesis is reasonable, without testing it (i.e., altering linker length and observing altered primer length), it is still a hypothesis. Some of the language in the manuscript (especially the abstract) is too

strong.

The proposed tape-measure mechanism is intriguing and should be explored more thoroughly. For example, if the hypothesis is correct then it would mean that shortening the linker would shorten the primer. However, this explanation is not clear because some of the sequences in Supplemental Figure 11 show linkers that are both longer and shorter than T4. However, the T4 primer is already quite short (5bp) and it is hard to imagine how it could get significantly shorter than 4bp, the length in T7 phage. What is the linker length in T7? Additionally, the Aquifex linker is by far the shortest of the sequences in Supp Fig11. (Note: I'm assuming that the sequence labeled Aaeo_DnaG is Aquifex aeolicus. This should be clarified in the ms.) It is hard to imagine that the primer of Aquifex is much shorter than in T4, especially considering the normal growth temperature of this thermophile. This would argue against the proposed mechanism.

More exploration of how this proposed mechanism fits into the broader context of phage and bacterial helicase/primase systems would be nice.

The reviewer was correct that "Aaeo" stands for "*Aquifex aeolicus*". We have clarified this and other bacterial names in the revised manuscript. The T7 phage primase is structurally different from the T4 primase; therefore, the primer length determinant may be different. The T7 primase is covalently linked to the helicase such that the stoichiometry between the primase and helicase is 1:1. The flexible T7 linker loop between primase and helicase is 13-amino-acid (residues 54-63) long, shorter than the 18-amino-acid linker loop (residues 97-115) of the T4 primase.

To address the reviewer's concern, we have performed additional experiments to test whether the linker loop contributes to the length of the primer synthesized by the T4 primase and added these results to the manuscript (**Supplementary Figure 17**). It turns out the reviewer is correct that the linker may not be the sole or major determinant of the primer length. We have softened the language in the abstract and added the following to the Results and Discussion sections:

In the RESULTS section: "*To test this hypothesis, a series of mutant primases was designed with either shortened or extended linker loops between the ZBD and RPD that included both conservative (e.g., the deletion or insertion of 1-2 residues) and substantial (e.g., the deletion or insertion of 4-5 residues) changes to the linker length, as well as the insertion of flexible (e.g., GGGGS sequence) or rigid (e.g., a repeat of the current linker sequence of PKEL) residues into the linker. The RNA primer synthesized by the WT primase and the 5 linker loop mutants was predominantly a pentamer ($93.9 \pm 0.9\%$) with very minor amounts of hexamer ($2.8 \pm 0.3\%$) or tetramer ($3.3 \pm 0.7\%$) RNA primers observed (Supplementary Fig. 17). Only in the case of mutant linker 2, where the linker loop was shortened by four residues, was an appreciable effect on the length of the primer noticed. While the predominant RNA primer was still a pentamer ($95.2 \pm 0.9\%$), the amount of hexamer primer ($1.2 \pm 0.1\%$) decreased and the amount of tetramer primer ($3.7 \pm 0.8\%$) increased with respect to the WT and other mutant primases as one would expect if the shorter linker loop restricted the rotation of the RPD thereby limiting the length of the primer that could be synthesized.*"

In the DISCUSSION section: "*Results from the series of mutant primases with altered linker loops demonstrated that it is possible to affect the length of the primer synthesized when the linker loop is shortened by four residues. The effect was subtle; slightly less hexameric primer and more tetrameric primer was synthesized, albeit the pentameric primer was still the predominant product. Therefore, other factors beside the length of the linker loop also influence the length of the primer that is synthesized by the T4 primase. One potential factor is the affinity*"

of the primase to the helicase. We suggest that a lengthening primer/template may exert an increasing force on the primase, and upon reaching the full length, the primase is pushed away and dissociates from the helicase, leading to the termination of primer synthesis. Such a scenario is consistent with the knowledge that T4 primase alone has little activity and is fully active only in complex with the helicase in the context of the primosome (Nossal, 1987 #58;Hinton, 1987 #59)"

Minor comments:

1) Figure 5d has hydrogen bonds that are hard to see versus the atoms.
We have changed the hydrogen bond color to black in **Fig. 5c-d**.

2) It was hard to find the linker helix and the red dashed box in figure Fig 2. Even when searching for them it took me a while to find them for some reason.
We have changed the color to yellow and increased the thickness of the dashed line in **Fig. 2**.

3) In the discussion, the authors write: "Indeed, the "scissor-like" motion into the nearly antiparallel 4-helical bundle configuration increases the interface and promotes compaction of the N-tier hairpin dimers. This binding energy driven-activation hypothesis is further supported by our observation that the mutant gp41 helicase, lacking ATP hydrolytic activity, adopted the active planar ring in the assembled primosome." I'm not following the logic between these two sentences. Please consider revising to make this connection more clear between ATPase activity and the scissoring of the helical bundles.

We apologize for the confusion. We meant that the conformational change from the open spiral to the closed ring is independent of ATP hydrolysis. This claim is supported by two observations: 1) the inactive helicase mutant lacking ATP-hydrolysis activity can still form the closed conformation, and 2) the additional interface of the NTD hairpin dimers formed in the closed state contributes to stabilizing this conformation or conversion from the inactive-to-active transition. We have clarified this: "*We suggest that the binding energy from two sources drives the inactive-to-active conformational transition: one source is the binding energy of the two additional DNA nucleotides inside the active helicase, and the other source is the binding energy from the "scissor-like" motion of the N-tier helical hairpin dimers that increases the interface and promotes compaction of the N-tier hairpin dimers. This binding-energy-driven transition hypothesis is supported by our observation that the mutant gp41 helicase, lacking ATP hydrolytic activity, also adopted the active planar ring in the assembled primosome*".

Reviewer #2 (Remarks to the Author):

In the manuscript by Madru et al., the authors present multiple cryo-EM structures showing how the T4 bacteriophage gp41 helicase interacts with ssDNA and the gp61 primase to form a 'primosome' complex. The structures are used to propose a molecular mechanism for primosome assembly, describe how the stoichiometry of helicase to primase is determined, and explain why RNA primers are limited to pentaribonucleotides in the T4 system. This manuscript provides several novel insights into the T4 primosome and compares the structures and proposed mechanisms with the equivalent prokaryotic/bacteriophage replication machinery. Overall, the work is clearly described and nicely illustrated, the reporting of the data and methodology are sufficiently detailed and transparent, and the structural work appears sound, however, the paper would benefit from experimental validation of some of the key conclusions.

My primary concern is that the study is purely structural, with no attempt to validate the hypotheses or mechanistic models proposed with complementary assays. In the clearest

example, a major claim of the manuscript is that one the structures—of the primase in the helicase mutant (E227Q) primosome—enables the authors to define the mechanism by which the RNA primer is limited to a pentaribonucleotide. The proposed mechanism is based on the length of the linker loop between the gp61 ZBD and RPD limiting the size of the primer, by limiting the movement of the RPD domain relative to the helicase/ZBD. Whilst this is an appealing model, the structural data alone is currently insufficient to support this conclusion. There is a clear improvement in the density for the linker in the mutant structure relative to the wild-type equivalent, additional RNA density can be seen for the RNA-DNA duplex and the RPD appears to shift slightly; however, the resolution of the maps around these key elements is limited (~5-7 Å or worse) and the density in some parts is weak (this is improved in the DeepEMhancer maps, but the fit of the RNA-DNA duplex into this map is relatively poor). Therefore, the exact positions of the RNA-DNA hybrid, surrounding residues and linker cannot be built with complete confidence. It is also notable that the linker that has currently been built into the model for the mutant primase does not appear to be fully extended. Given the above, the manuscript would benefit significantly from additional evidence to validate the proposed model, importance of the linker length and the relevance of any amino acids/interactions that are proposed to be key for determining the RNA primer length. For example, have the authors investigated whether the linker length can be modified and thereby change the length of the primers generated in a biochemical assay?

We appreciate the reviewer's insightful comments. Reviewer #1 had a similar concern. Please see our response to Reviewer #1 regarding additional experiments we have performed to test the hypothesis that the linker loop contributes to defining the length of the primer (new **Supplemental Figure 17**). Briefly, we found the primer length is only slightly changed by modifying (lengthening or shortening) the linker length. We have therefore softened our claim on the importance of the linker length. In the Discussion section, we further suggest that the affinity between the primase and the helicase may also be an important factor in determining the primer length.

Minor points

It is unclear why the authors chose a substrate that is centrally primed, with 30 and 35 nt of ssDNA on either side. Given the DNA seen spanning the helicase channel is only 12 nt, is it possible that the helicase could bind to the DNA with the primer either in front or behind, complicating interpretation of the data? Further explanation of the design of the experiment would aid interpretation.

We agree with the reviewer that – in hindsight – we could have used a shorter templated DNA. However, prior to our cryo-EM studies, we did not know the required size of ssDNA substrate to form a stable primosome. We also did not know the length of the template DNA spanning the central channel of the helicase or which region of the template DNA traverses the active site of the primase. However, our previous work showed that a 45-nt ssDNA with a priming recognition site was required for detectable priming activity¹ and that a longer ssDNA substrate might be required to accommodate a DNA loop between the helicase and primase². Therefore, we used in our study a relatively long template DNA knowing that we could computationally separate and classify the in vitro assembled particles and identify the primosome if it formed. We have added these explanations to the first paragraph of the Result section.

We also agree with the reviewer that the helicase can binds our long template DNA either in front of the primer (3' end of the template) or in the back of primer (5' end of the template). However, this is not a concern in our goal of assembling a primosome, because we know that

helicase binding on ssDNA is unidirectional, with the 3' end at the C-terminal ATPase tier and the 5' end at the N-terminal tier (see review by O'Donnell and Li, NSMB 2018, PMID: 29379175). Therefore, only the helicase bound in front of the primer (3' end of the template) can bind the primase that would need to also bind to the primer. And the helicase bound in the back of primer (5' end of the template) is unable to bind the primase to assemble a primosome. Indeed, our 3D classification did reveal a helicase-only subpopulation (**Supplemental Figure 4**, the middle 3D class in blue, 16% of the particle population), which we suspect were those helicases that bound in the back of the primer and were unable to bind the primase to form a primosome.

The authors appear to only detect one state of the primase bound to the mutant helicase complex, rather than the three states observed for the WT complex in the presence of ATP γ S, does this tell us anything about primase recruitment or helicase translocation? For example, in the absence of ATP hydrolysis, the binding state may represent initial primase recruitment prior to any helicase translocation. Is the primase found binding across the helicase seam, where it may stabilize the closed form?

We apologize for the confusion. We did observe primase binding in multiple orientations relative to the helicase hexamer in the mutant primosome. However, due to the smaller number of particles with primase density, we chose to combine all particles with a bound primase to generate a primosome map with averaged helicase density but with a better-defined primase density (**Supplementary Fig. 14**). And yes, we indeed observed only primase bound to the closed-state helicase, and we believe that the primase has stabilized the helicase hexamer in the closed state. We have revised the Method section (Image process sub section) pertaining to the mutant primosome 3D reconstruction.

The deposited structure of the mutant helicase complex (PDB: 8DWJ) only includes the primase and DNA-RNA duplex but lacks the entire helicase. Given the structure is based on a map of the entire complex, it would be much more useful to other researchers to deposit a structure for that complete complex. Presumably, completing this model would be relatively trivial and would be more informative. Additionally, Table 1 includes the global resolution of 3.9 Å for the entire map, but this is not the resolution that represents the density for which the model of the primase structure was built.

We previously did not build atomic model for the entire mutant primosome because the helicase region is an average of multiple conformations -- in order to obtain a decent EM map for the primase (see our response to the preceding question). However, we agree with reviewer's suggestion and have now deposited the EM map from the focus-refined primase region at 4.1-Å resolution, the EM map from focus-refined helicase hexamer at 3.6-Å resolution, and the composite EM map of the mutant primosome. We have also built atomic model for the mutant primosome and deposited it in the PDB bank (PDB ID 8GAO; see revised Supplemental Table 1). We have explained this in the revised Method section.

Page 7, final paragraph "We found that about 18% of primosome particles had gp61 primase flexibly associated..." does this refer to the "16%" class shown in Sup Fig. 4? If so, the numbers should be consistent. It is also unclear from the current manuscript how the authors know that these particles are primase associated?

We thank the reviewer for pointing out the issue. The number in the text should be 16%, and we have made the change. The calculation is detailed here: The closed-ring helicase hexamer particles (100%) were classified into three classes; two of which have electron density for the

location of primase (84% combined) and one class with no density for primase bound (16%). From the 84% particles with primase bound, we were able to further classify these particles into primase pose 1 – 3 (26.2%, 14.2% and 22.4%, respectively) and primase with only the RPD domain bound to the helicase (4.9%). Combined, these classes accounted for 67.7% in the 84% of the particles with electron density for primase. Therefore, we assumed the remaining 16.3% of the particles have primase flexibly bound to the closed helicase hexamer such that no specific orientation could be determined. We have added this explanation/calculation in the Method section pertaining the wild type primosome image process.

Page 9, “The gp61 primase region achieved ~4.0 Å local resolution” – this appears to conflict with the local resolution maps displayed in Sup. Fig. 14d which suggests 5-7 Å and possibly lower resolution. This figure should also be amended so that the resolution colour scale extends beyond the lowest resolution limit so that the true limits of the resolution can be evaluated.

Thank you for pointing out this discrepancy. We performed another round of local refinement on the primase region with cryosparc³; the resolution of this map reached 4.1 Å. The map was deposited, and the local resolution distribution was added in revision as **Supplementary Figure 15e**.

Page 11, top “we suggest that that the transition is driven by the binding energy of the two additional DNA nucleotides from 10 nt bound in the inactive spiral form, to 12 nt bound in the active ring.” Is it possible to form the closed ring on shorter DNA constructs to test this (perhaps with the helicase-dead mutant)? Negative stain EM may be sufficient for distinguishing between open and closed on different length ssDNA substrates?

Thanks for pointing out this misstatement. Compared with the large conformational movement, it is unlikely that additional nucleotides binding can drive the large conformational change. We have modified the sentence to read “*Therefore, we suggest that the binding energy of the N-tier helical hairpin dimers post the “scissor-like” motion drives the inactive-to-active conformational transition. And the binding energy of the two additional DNA nucleotides inside the active helicase further stabilizes the active configuration. This binding-energy-driven transition hypothesis is supported by our observation that the mutant gp41 helicase, lacking ATP hydrolytic activity, also adopted the active planar ring in the assembled primosome*”.

The parameters used for DeepEMhancer map modification should be provided in the methods section and where DeepEMhancer maps are used in figures, this should be noted in the figure legends.

The parameters have been included in the revised Method section and in the corresponding revised figure legends.

Fig. 1c legend – The resolution value (3.2 Å) given for primase in the composite map of structure 3-II is misleading, that estimate is largely based on the helicase portion that is not shown. The local resolution of the primase region being described is significantly lower than 3.2 Å.

We did indicate that this is an overall resolution. However, we agree with the reviewer we should be more specific. We have now added an explanation regarding the resolution of map 3-II in the revised figure legend.

Fig. 6a/b – Transparent surfaces shown with the fitted model would better allow interpretation of the model and fit to the density.

We thank the reviewer for this suggestion. We have incorporated the changes in the revised figure and legend.

Fig. 7. To avoid misleading a non-expert reader, it may help to highlight in the figure and/or legend that helicase loading would normally be performed by the helicase loader gp59 in vivo. We thank the reviewer for the good suggestion. We have modified the figure legend to add this detail.

Reviewer #3 (Remarks to the Author):

The work by Feng et al. presents several cryo-EM structures of primosome complex formation from the T4 phage system. Using snapshots of a sequential series of structural intermediates, the authors develop a molecular framework for describing the formation of an active helicase/primase complex bound to a primer-template DNA intermediate. The structures show how DNA induces conformational changes that lead to the closing of the gp41 helicase ring. This closure exposes a cryptic hydrophobic site, consequently leading to the recruitment of the primase.

Overall, the work is of high technical quality and the proposed mechanism provides fresh and long-awaited insights into the ingenious control mechanisms that regulate both helicase-DNA interactions and primase/helicase complex formation. The researchers also provide a intriguing and plausible hypothesis about how primer length is controlled based on the observation of two different gp61 conformations and the ability of a linker loop that connects the ZBD and the RPD of the primase to stretch to a predefined limit. Given its breadth and significance, the work should be published in Nature Communications pending some minor revisions.

Primary comments

- The authors hypothesize that the physical constraint of the linker loop length between the RPD and ZBD of gp61 primase dictates the primase counting. It follows that changing this linker length can influence the size of the primer synthesized. It would be important to test biochemically whether changing the linker length indeed affects primer length.

Thanks for the insightful comments. The other reviewers had similar concern regarding additional experiments to test the linker function. Please see above our response to Reviewer #1. Briefly, we found the primer length is only slightly changed by modifying (lengthening or shortening) the linker length. We have therefore softened our claim on the importance of the linker length. In the Discussion section, we further suggest that the affinity between the primase and the helicase may also be an important factor in determining the primer length.

- The authors observe a 6:1 stoichiometry for the helicase: primase complex, which (as they note) has been suggested before. However, other studies have reported evidence for a greater number of primases binding to the helicase (e.g., Yang et al, JBC, 2005). Since three equivalent binding sites are available for the primase to bind the helicase, can the authors comment on the structural basis for their observed 6:1 stoichiometry and why it would appear to discount the existence of multi-primase binding events?

We concede that the current research was not designed to address the issue of stoichiometry. However, our structure appears to be consistent with 6:1 due to the bipartite binding of the primase. In other words, one primase occupies two of the three available binding sites on

helicase (one site by primase ZBD and the second site by primase RPD), so there is only one spare site that is insufficient for a stable (productive) binding of a second primase. We have added the following statement to the manuscript: “*The stoichiometry of the T4 primosome has been controversial, with reports ranging from 6:1^{4,5} to 6:6^{1,6} helicase to primase subunits in the complex. The bipartite binding mode of gp61 primase to gp41 helicase observed in the majority of the primosome particles is consistent with a stoichiometry of 6 helicase:1 primase. But static structures can underestimate the complexity of protein quaternary structures and/or miss the dynamic equilibrium between different quaternary forms of complexes in solution⁷. The observation of subpopulations of primosome particles with other than bipartite or undeterminable primase binding modes to helicase suggests that other than 6:1 or flexible stoichiometries are also possible for the T4 primosome in solution*”.

- If one aligns on the axis of the bound DNA, then either the gp41 subunits must adopt a spiral configuration to track along the backbone (which appears to have a helical rise of some 25-30 Å or so), or the L1/L2 loops must adjust their vertical position to maintain comparable contacts with the substrate. Which is it? The answer bears on whether it makes sense to call the closed gp41 ring planar, or whether this is a matter of how the helicase ring is tilted in the perspective of the figures with respect to the bound DNA.

We apologize for the confusion and would like to clarify this matter. The gp41 L1/L2 loops do not raise vertically relative to their respective CTDs, rather, the gp41 CTDs are arranged with a small helical rise. However, the step size of the helical rise in the closed-ring state is much smaller than the open spiral helicase hexamer or the homologous bacterial DnaB helicase (side-by-side comparison shown in **Supplementary Figure 8**). Therefore, the active closed-ring T4 gp41 helicase is largely a “planar ring”, although it is not strictly planar. We have labeled it “planar” to distinguish it from the open spiral and the previous DnaB spiral structures.

Minor points

- Figs. 3f and 3d are called out before Figs. 3a-c. Please reorder panels and adjust callouts. We have reordered the panels of Figure 3.
- Fig. 6d and associated text – Lys51 appears to be an Arg in the figure? Please clarify. Thanks for catching this error; it should be Arg51. We have corrected the figure.
- Fig. SF10c-e. Please show the ssDNA and ssRNA as sticks rather than cartoon to aid in visualizing the fit to density. The nucleotides in the density maps in SF10c-e are now shown as sticks as suggested.

Cited References:

- 1 Valentine, A. M., Ishmael, F. T., Shier, V. K. & Benkovic, S. J. A zinc ribbon protein in DNA replication: primer synthesis and macromolecular interactions by the bacteriophage T4 primase. *Biochemistry* **40**, 15074-15085, doi:10.1021/bi0108554 (2001).
- 2 Manosas, M., Spiering, M. M., Zhuang, Z., Benkovic, S. J. & Croquette, V. Coupling DNA unwinding activity with primer synthesis in the bacteriophage T4 primosome. *Nat Chem Biol* **5**, 904-912, doi:10.1038/nchembio.236 (2009).
- 3 Punjani, A., Rubinstein, J. L., Fleet, D. J. & Brubaker, M. A. cryoSPARC: algorithms for rapid unsupervised cryo-EM structure determination. *Nat Methods* **14**, 290-296, doi:10.1038/nmeth.4169 (2017).

- 4 Jing, D. H., Dong, F., Latham, G. J. & von Hippel, P. H. Interactions of bacteriophage T4-coded primase (gp61) with the T4 replication helicase (gp41) and DNA in primosome formation. *J Biol Chem* **274**, 27287-27298, doi:10.1074/jbc.274.38.27287 (1999).
- 5 Jose, D., Weitzel, S. E., Jing, D. & von Hippel, P. H. Assembly and subunit stoichiometry of the functional helicase-primase (primosome) complex of bacteriophage T4. *Proc Natl Acad Sci U S A* **109**, 13596-13601, doi:10.1073/pnas.1210040109 (2012).
- 6 Yang, J., Xi, J., Zhuang, Z. & Benkovic, S. J. The oligomeric T4 primase is the functional form during replication. *J Biol Chem* **280**, 25416-25423, doi:10.1074/jbc.M501847200 (2005).
- 7 Marciano, S. *et al.* Protein quaternary structures in solution are a mixture of multiple forms. *Chem Sci* **13**, 11680-11695, doi:10.1039/d2sc02794a (2022).

REVIEWERS' COMMENTS

Reviewer #1 (Remarks to the Author):

The authors have done a great job responding to the reviewers initial comments. In particular, I'm very happy that the authors included experiments examining primer length in response to modification of the linker length. The paper should be published.

One thing that I would urge the authors to critically examine their hypothesis that linker length controls primer length. I feel that the authors did an admirable job of testing this hypothesis experimentally, with multiple variants that lengthen or shorten the linker. Because none of these mutants had much of an effect on primer length (the authors make a point about one of the mutants, but the changes in length are not particularly substantial), I would argue that their experiments don't support their hypothesis.

I certainly think that the hypothesis was a reasonable one to propose based on the structure, but I would say with the lack of experimental support, the hypothesis is unlikely to be true. So I would urge the authors to reexamine how they frame their results in the abstract and elsewhere in the paper so as not to overinterpret their results.

Other than that, this is an important piece of work and should be published asap.

Reviewer #2 (Remarks to the Author):

The authors have improved the manuscript and addressed most of my original concerns. I only have a few minor outstanding concerns relating to the proposed "measuring tape" mechanism to regulate primer length, as listed below:

Sup Fig. 17 shows the results of the experiment to investigate priming by mutant primases and plots showing the 'priming rate', however, values for the priming rates are not shown and the results section does not refer to rates, but rather to the proportions of primers at different lengths, presumably at the end point of the reaction. Therefore, the figure does not correspond with the data discussed. If the authors would like to compare the rates, then these should be discussed in the results section, the rates calculated should be shown and the method for calculating them described in the methods. If they would like to discuss end-point ratios of different primer lengths, they should show the comparison of these ratios in the figure with error bars (it is unclear how the errors quoted in the results section were calculated, or how many times the experiment was repeated) and show whether any differences are statistically significant.

The data the authors now include, indicates that the linker length has very minimal or no impact on defining the pentameric primer length (error estimates and significance of differences may help determine which). Whilst the only mutant to show an influence on primer length (L2) appears to form the pentameric primer more efficiently than the native linker, lengthening the linker appears to have no impact on the primer length. In my view, the work uses structural data to propose a feasible hypothesis that the linker length is a major determinant of primer length, tests this hypothesis and can then reject it. This is a valid finding.

Whilst the description of these experiments in the results and discussion sections are balanced and reasonable (providing the differences are statistically significant), the line "we suggest that the linker loop between the gp61 ZBD and RPD contributes to the T4 pentaribonucleotide primer" in the abstract feels misleading (even if true in a strict sense), because it gives unnecessary emphasis to the very

minor role the linker plays, when other much more significant mechanisms are clearly required.

Reviewer #3 (Remarks to the Author):

Authors have satisfied all the objections raised from my end. This manuscript can be published in Nature Communications journal as such.

RESPONSE TO REVIEWERS' COMMENTS

Reviewer #1 (Remarks to the Author):

The authors have done a great job responding to the reviewers initial comments. In particular, I'm very happy that the authors included experiments examining primer length in response to modification of the linker length. The paper should be published.

One thing that I would urge the authors to critically examine their hypothesis that linker length controls primer length. I feel that the authors did an admirable job of testing this hypothesis experimentally, with multiple variants that lengthen or shorten the linker. Because none of these mutants had much of an effect on primer length (the authors make a point about one of the mutants, but the changes in length are not particularly substantial), I would argue that their experiments don't support their hypothesis.

I certainly think that the hypothesis was a reasonable one to propose based on the structure, but I would say with the lack of experimental support, the hypothesis is unlikely to be true. So I would urge the authors to reexamine how they frame their results in the abstract and elsewhere in the paper so as not to overinterpret their results.

We have revised the manuscript to reflect the limited to no role of the linker loop on the length of primer synthesized by the T4 primase. Specifically, we made the following changes:

- In the Abstract, we deleted the following sentence: "Based on two observed primosome conformations – one in a DNA-scanning mode and the other in a post RNA primer-synthesis mode – we suggest that the linker loop between the gp61 ZBD and RPD contributes to the T4 pentaribonucleotide primer."
- In the Results, we reported the small effect of Linker 2 on the synthesized primer as "The RNA primer synthesized by the WT primase and four of the five linker loop mutants was predominantly a pentamer ($86 \pm 1\%$) with minor amounts of hexamer ($2.7 \pm 0.4\%$), tetramer ($2.2 \pm 0.4\%$), and trimer ($9 \pm 1\%$) RNA primers observed (Supplementary Fig. 17). Only in the case of mutant linker 2, where the linker loop was shortened by four residues, was **a small but** appreciable effect on the length of the primer noticed. While the predominant RNA primer was still a pentamer, the amount of pentamer ($82.4 \pm 0.6\%$) and hexamer ($1.0 \pm 0.1\%$) primer decreased slightly and the amount of trimer primer ($13.9 \pm 0.1\%$) increased with respect to the WT and other mutant primases as one **might** expect if the shorter linker loop restricted the rotation of the RPD thereby limiting the length of the primer that could be synthesized."
- In the Discussion, we down played the role of the linker loop on the primer length by writing: "We **hypothesized** that the linker loop length could contribute to the "measuring tape" function of the gp61 primase that limits the primer to 5 nt in the T4 system. However, shortening the linker loop by four residues had only a subtle effect on the length of the synthesized primer producing slightly less pentamer and hexamer primers and more trimer primers. Therefore, other factors besides the linker loop also likely influence the length of the primer synthesized by the T4 primase. One potential factor is the affinity of the primase to the helicase. The elongating ssDNA emerging from the gp41 helicase may apply increasing pressure on the primase to dissociate from the helicase, leading to the termination of primer synthesis."

Other than that, this is an important piece of work and should be published asap.

Reviewer #2 (Remarks to the Author):

The authors have improved the manuscript and addressed most of my original concerns. I only have a few minor outstanding concerns relating to the proposed “measuring tape” mechanism to regulate primer length, as listed below:

Sup Fig. 17 shows the results of the experiment to investigate priming by mutant primases and plots showing the ‘priming rate’, however, values for the priming rates are not shown and the results section does not refer to rates, but rather to the proportions of primers at different lengths, presumably at the end point of the reaction. Therefore, the figure does not correspond with the data discussed. If the authors would like to compare the rates, then these should be discussed in the results section, the rates calculated should be shown and the method for calculating them described in the methods. If they would like to discuss end-point ratios of different primer lengths, they should show the comparison of these ratios in the figure with error bars (it is unclear how the errors quoted in the results section were calculated, or how many times the experiment was repeated) and show whether any differences are statistically significant.

We agree with the reviewers comments and modified Sup. Fig. 17 in the following ways:

- Panel b) shows a graph of the pentamer primers synthesized by WT and each gp61 primase linker loop mutant versus time to demonstrate that the reaction is linear over the 4 min assay time.
- Panel c) shows a dot plot of the total primers synthesized by WT and each gp61 primase linker loop mutant versus time to demonstrate that the overall priming activity of the primase mutants was not affected by the changes made to shorten or extend the linker loops.
- Panel d) shows a dot plot compiled from triplicate priming assays depicting the percentage of various length primers synthesized by the indicated WT and linker loop mutant primases after 4 min. These are the proportions of various length primers discussed in the text.

The data the authors now include, indicates that the linker length has very minimal or no impact on defining the pentameric primer length (error estimates and significance of differences may help determine which). Whilst the only mutant to show an influence on primer length (L2) appears to form the pentameric primer more efficiently than the native linker, lengthening the linker appears to have no impact on the primer length. In my view, the work uses structural data to propose a feasible hypothesis that the linker length is a major determinant of primer length, tests this hypothesis and can then reject it. This is a valid finding.

Whilst the description of these experiments in the results and discussion sections are balanced and reasonable (providing the differences are statistically significant), the line “we suggest that the linker loop between the gp61 ZBD and RPD contributes to the T4 pentaribonucleotide primer” in the abstract feels misleading (even if true in a strict sense), because it gives unnecessary emphasis to the very minor role the linker plays, when other much more significant mechanisms are clearly required.

Please see the detailed response to Reviewer #1 on similar comments.

Reviewer #3 (Remarks to the Author):

Authors have satisfied all the objections raised from my end. This manuscript can be published in Nature Communications journal as such.